# COMPASS: TRAINING-FREE GUIDANCE FOR SKILL DISCOVERY WITH HUMAN FEEDBACK

## ABSTRACT

Unsupervised skill discovery (USD) aims to learn diverse behaviors without reward functions, but often results in task-irrelevant or hazardous behaviors due to uniform exploration. Guided skill discovery (GSD) addresses this issue by incorporating human intent to focus exploration on meaningful and safe regions. However, existing GSD methods typically rely on pre-defined rules, expert demonstrations, or training instruction models, which are either costly to obtain or ineffective with sparse human feedback. To tackle this, we identify a key insight: a semantically coherent skill latent space, where nearby embeddings correspond to behaviors with similar human desirability, enables training-free guidance from sparse feedback. Building on this insight, we propose COMPASS, a **training-free** GSD framework that ensures semantic coherence in the latent space. Exploiting the coherence of this latent space, COMPASS constructs a dense guidance signal in a training-free manner in this latent space, eliminating the need for any model training beyond the skill policy itself. This guidance signal is then integrated into skill discovery objectives to direct exploration toward human-desirable regions. Theoretical analysis guarantees the reliability of our training-free guidance signal, and extensive experiments across diverse state-based and pixel-based tasks show that COMPASS learns diverse, human-aligned skills, avoids hazardous behaviors, and achieves superior downstream performance with minimal human feedback.

## 1 INTRODUCTION

Unsupervised learning aims to learn meaningful representations or behaviors through self-supervised objectives without pre-defined task-specific goals, which has shown effectiveness across domains, such as computer vision (Chen et al., 2020; Radford et al., 2021) and natural language processing (Devlin et al., 2019; Brown et al., 2020). In reinforcement learning, *unsupervised skill discovery* (USD) builds on this idea to learn diverse, distinguishable behaviors that broadly cover the state space, facilitating downstream tasks. However, USD's uniform exploration strategy often leads to useless or harmful skills (Kim et al., 2023), especially in complex scenarios, where vast state spaces include irrelevant or hazardous regions. This inefficiency not only wastes computational resources but also limits USD's practical applicability in real-world tasks.

To address the limitations of USD, *guided skill discovery* (GSD) methods draw inspiration from human cognition, where humans prioritize exploring potentially useful regions, rather than uniformly covering the state space (Du et al., 2023). By incorporating external human intent, GSD focuses exploration on meaningful and safe areas. However, existing GSD methods often ① rely on pre-defined rules or expert demonstrations (Kim et al., 2023; 2024), which can be costly and challenging to obtain in complex environments, and ② require training auxiliary models to encode human intent (Klemsdal et al., 2021; Kim et al., 2024), which risk overfitting with limited human feedback, leading to unreliable guidance in complex scenarios.

In this paper, we present a key insight: The latent skill space learned by USD is a powerful yet under-utilized resource for guidance, especially when such a latent space is *semantically coherent*, i.e., nearby embeddings correspond to behaviors with similar human desirability. While not inherent to all USD methods, such coherence can be achieved by aligning Euclidean distances in the latent space with the temporal distance (Kaelbling, 1993) in the state space, which ensures the continuity of human desirability within the latent space. Leveraging this property, human knowledge can be

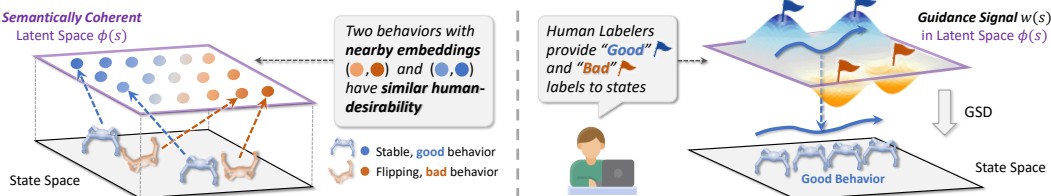

Figure 1: Overview of COMPASS. (1) We leverage a ***semantically coherent*** latent space of USD, where states with nearby embeddings share similar human desirability. (2) Using sparse human "good/bad" labels on states, COMPASS constructs a dense, training-free guidance signal $w(s)$.

efficiently injected: By assigning simple "good" or "bad" labels to a minimal number of states, we can construct a dense and consistent guidance signal, $w(s)$, across the state space in a theoretically grounded manner. This enables a new ***training-free*** guidance paradigm, eliminating the need for costly expert data or additional model training in prior work.

Based on this insight, we propose COMPASS, a training-free GSD framework that leverages the semantic coherence of the skill latent space. First, COMPASS enforces semantic coherence by constraining the latent embeddings of temporally adjacent states to be close. Then, leveraging the semantic coherence property, it constructs a dense guidance signal $w(s)$ from sparse human feedback, by propagating the semantics of "good" and "bad" labeled states within the coherent latent space. Theoretical analysis confirms this guidance signal effectively prioritizes human-desired behaviors. To ensure the guidance signal's accuracy, which requires labeled states to cover the state space sufficiently, COMPASS employs an active query strategy to select under-explored behaviors for human labeling. By integrating the guidance signal into the USD's intrinsic reward, COMPASS directs exploration toward desirable regions without requiring expert data or auxiliary model training.

Both theoretical analysis and experimental results validate the effectiveness of COMPASS. Theoretically, COMPASS enables effective, training-free guidance signal construction. Empirically, across various complex tasks, COMPASS learns diverse, human-aligned skills with minimal feedback, avoiding hazards, focusing on task-relevant regions, and enhancing downstream performance. Visualizations of learned skills further confirm that COMPASS enables safe and meaningful exploration.

In summary, our contributions are threefold:

- We propose a novel training-free paradigm for GSD: By enforcing semantic coherence in the latent space, human intent can be directly injected to guide exploration via this latent space. This eliminates the need for auxiliary models or expert demonstrations in prior works, and establishes a new framework for human-aligned exploration.

- We implement this paradigm in COMPASS, a training-free method that leverages human-defined "good/bad" labels to construct a guidance signal. This method is simple, effective, theoretically grounded, and provides provable error guarantees.

- Extensive experiments on both state-based and pixel-based tasks show that COMPASS learns diverse, safe, and human-aligned skills with minimal human feedback, while achieving downstream task performance close to oracle-level guidance, showing its effectiveness.

## 2 PRELIMINARIES

**Unsupervised Skill Discovery (USD).** Unsupervised reinforcement learning considers a Markov decision process (MDP) without reward functions, which is characterized by $(\mathcal{S}, \mathcal{A}, \mathcal{P}, \mu_0)$, where $\mathcal{S}$ and $\mathcal{A}$ are the state and action spaces, $\mathcal{P} : \mathcal{S} \times \mathcal{A} \to \Delta(S)$, and $\mu_0$ is the initial state distribution.

USD methods aim to acquire knowledge about the environment by learning a set of distinguishable skills that collectively cover the state space. This is achieved by introducing a latent skill space $\mathcal{Z}$ and training a skill-conditioned policy $\pi : \mathcal{S} \times \mathcal{Z} \to \Delta(\mathcal{A})$. During training, skills are sampled from a prior distribution $p(z)$, and the agent interacts with the environment using the policy $\pi(a|s, z)$ based on the sampled skill. Once learned, these skills can be reused to facilitate downstream tasks,

such as (1) learning a hierarchical policy that uses skill-conditioned policies as lower-level policies, or (2) selecting skills that maximize the task reward in a zero-shot manner.

**Distance-maximizing Skill Discovery (DSD).**  A common approach to USD is to maximize the mutual information (MI) $I(s, z)$ between skills $z$ and the visited states $s$ (Eysenbach et al., 2019). However, this can lead to static behaviors, as MI only ensures skill discriminability without encouraging broader state coverage (Park et al., 2023b). DSD methods (Park et al., 2023a; 2022b; 2023b) address this by aligning state space distances to latent skill space distances. A typical objective is:

$$\sup_{\pi,\phi} \mathbb{E}_{\tau,z} \left[ \sum_{t=0}^{T-1} (\phi(s_{t+1}) - \phi(s_t))^\top z \right] \text{ s.t. } \|\phi(x) - \phi(y)\|_2 \le d(x, y), \ \forall (x, y) \in \mathcal{S}, \qquad (1)$$

where $\tau = (s_0, \dots, s_T)$ denotes the trajectory, $d(\cdot, \cdot)$ is a distance metric in the state space. This objective can be optimized via dual gradient descent (Boyd & Vandenberghe, 2004) with a Lagrange multiplier $\lambda$, and the policy is updated with intrinsic reward $r(s, z, s') = (\phi(s') - \phi(s))^\top z$.

**Guided Skill Discovery (GSD).**  In complex scenarios, USD can be inefficient, as many of the learned skills may be irrelevant or even harmful to downstream tasks (Kim et al., 2024). GSD addresses this issue by incorporating human intent to guide skill learning toward desirable behaviors while avoiding undesirable ones. A typical objective is:

$$\sup_{\pi,\phi} J_{\text{USD}}(\pi, \phi) + \lambda_{\text{guide}} \cdot J_{\text{guide}}(\pi, \phi) \quad \text{s.t.} \quad C_{\text{USD}}(\phi) \le 0, \quad \lambda_{\text{guide}}^c \cdot C_{\text{guide}}(\phi) \le 0, \qquad (2)$$

where $J_{\text{USD}}(\pi, \phi)$ represents the USD objective, such as mutual information $I(s, z)$ or the DSD objective. $C_{\text{USD}}(\phi)$ denotes constraints in USD objectives, such as those in Eq. 1. $J_{\text{guide}}(\pi, \phi)$ reflects the guidance objective derived from human intent, which may include expert trajectories (Kim et al., 2024; Klemsdal et al., 2021) or pairwise human preferences (Hussonnois et al., 2023; 2025). $C_{\text{guide}}(\phi)$ is the guidance in the form of constraints, like analytical constraint formulas for safety (Kim et al., 2023). The coefficients $\lambda_{\text{guide}}, \lambda_{\text{guide}}^c \ge 0$ adjust the strength of guidance.

**Human Feedback Format.**  We consider an interactive human-in-the-loop setting, where a labeler evaluates agent behaviors by providing feedback on state sequence segments $\sigma = (s_t, \dots, s_{t+H})$ of fixed length $H$, also referred to as "queries" in this paper. Each segment is assigned a scalar label $y \in \{0, 1, 2\}$, where $y = 2$ denotes a "good" segment (e.g., moving toward a goal), $y = 0$ denotes a "bad" segment (e.g., entering a hazardous area), and $y = 1$ denotes a "neutral" segment (e.g., neither contributing to task goals nor incurring risk). We assume all states within a segment share the same label. Labeled states are stored in a dataset $\mathcal{D} = \{(s, y)\}$, which is partitioned into subsets $\mathcal{D}_0$, $\mathcal{D}_1$, and $\mathcal{D}_2$ for bad, neutral, and good states, respectively.

## 3 COMPASS: TRAINING-FREE GUIDANCE FOR SKILL DISCOVERY

In this section, we propose COMPASS, a training-free guided skill discovery method, as illustrated in Fig. 1 and Algorithm 2. Section 3.1 presents our GSD framework, which employs a guidance signal $w(s)$ to direct exploration toward desirable regions. Section 3.2 constructs the guidance signal $w(s)$ in a training-free manner, by enforcing a semantically coherent latent space, and inferring $w(s)$ from sparse human labels within this space. As the guidance signal's accuracy relies on sufficiently state space coverage of labeled states, Section 3.3 introduces an active query selection mechanism that prioritizes under-explored behaviors for human labeling.

### 3.1 GSD FRAMEWORK WITH THE GUIDANCE SIGNAL

To enable training-free guidance, which propagates sparse human feedback semantics to create a dense signal without auxiliary model training, we require a latent skill space that meaningfully reflects the structure of the state space. We therefore build upon the DSD framework (Section 2), which learns a latent space $\phi(s)$ constrained to reflect state-space distances, and learns skills to maximize the distance traveled within the latent space. To incorporate human intent, we introduce

---

**Algorithm 1** COMPASS

---

**Require:** Feedback frequency $K$, total feedback number $N_{\text{total}}$, number of queries per feedback session $M$, total epoch number $T^{\text{e}}$

1: Initialize replay buffer $\mathcal{B}$, feedback buffer $\mathcal{D}_0, \mathcal{D}_1, \mathcal{D}_2$
2: **for** each epoch $e = 1, 2 \dots, T^{\text{e}}$ **do**
3:     Sample skill $z \sim p(z)$, rollout with policy $\pi(a|s, z)$ and store $(s, a, s')$ into $\mathcal{B}$
4:     **if** epoch % $K = 0$ and $|\mathcal{D}_0| + |\mathcal{D}_1| + |\mathcal{D}_2| < N_{\text{total}}$ **then**
5:         Select segments $\{\sigma_i\}_{i=1}^M \sim \mathcal{B}$ using the method in Section 3.3 and Algorithm 3
6:         Query labelers for feedback $\{y_i\}_{i=1}^M$, $y_i \in \{0, 1, 2\}$
7:         Save labeled states into feedback buffer, $\mathcal{D}_y \leftarrow \mathcal{D}_y \cup \{s : s \in \sigma_i, y_i = y\}_{i=1}^M$, $y = 0, 1, 2$
8:     **end if**
9:     Sample transitions from $\mathcal{B}$
10:    Calculate the guidance signal $w(s)$ with Eq. 7 and the smooth mechanism in Section 3.4
11:    Update the skill latent $\phi(s)$ with Eq. 11
12:    Update the Lagrange multipler $\lambda$ with Eq. 12
13:    Update the skill conditioned policy $\pi(a|s, z)$ with Eq. 13
14: **end for**

---

a guidance signal $w(s) : \mathcal{S} \to \mathbb{R}^+$ as a distance modifier. Formally, the DSD objective (Eq. 1) is extended as:

$$\sup_{\pi, \phi} \mathbb{E}_{\tau, z} \left[ \sum_{t=0}^{T-1} (\phi(s_{t+1}) - \phi(s_t))^\top z \right] \text{ s.t. } \|\phi(x) - \phi(y)\|_2 \leq w(x)d(x, y), \ \forall x, y \in \mathcal{S}, \quad (3)$$

This formulation captures a key intuition: assigning large $w(s)$ to human-desirable states relaxes the constraint on the latent space $\phi$, allowing for broader exploration in those areas. Conversely, small $w(s)$ in undesirable regions tightens the constraint, discouraging exploration. In essence, if we can construct a $w(s)$ that aligns with human intent, this framework naturally leads to human-desirable skills, as validated in prior works (Kim et al., 2024). Consequently, the complexity of GSD is reduced to constructing an effective, training-free guidance signal $w(s)$, which we discuss below.

### 3.2 TRAINING-FREE GUIDANCE IN SEMANTICALLY COHERENT LATENT SPACE

**Semantically coherent latent space.** Having established the GSD framework, we now address the prerequisites for constructing the guidance signal $w(s)$ in a training-free manner. Our key insight is that if the latent space $\phi(s)$ is *semantically coherent*, whereby nearby embeddings correspond to states with similar human desirability, a dense $w(s)$ can be inferred by propagating the semantics of sparse human-labeled states within the latent space.

We formalize the concept of *semantic coherence* as follows. Let $g(s) : \mathcal{S} \to \{0, 1, 2\}$ denote the human desirability for state $s$, with higher values indicating more desirable states. For a latent space $\mathcal{U}$ defined by $u = \phi(s) : \mathcal{S} \to \mathcal{U}$, we say $\mathcal{U}$ is semantically coherent, if $\forall \epsilon > 0, \exists \delta > 0$, such that

$$\|\phi(s_1) - \phi(s_2)\|_2 \leq \delta \implies P[g(s_1) = g(s_2)] \geq 1 - \epsilon \qquad \forall s_1, s_2 \in \mathcal{S}. \quad (4)$$

We would like to note that constructing a semantically coherent latent space is essential because the raw state space is typically not semantically coherent. For example, in robotic locomotion, a robot at a given $(x, y)$ position can be in either a stable (human-desirable) or fallen (undesirable) state. Although these states may be very close in the state space, e.g., differing only in joint angles or orientation, they exhibit completely different human desirability. In these scenarios, Euclidean distance becomes an inadequate measure of semantic similarity (Jiang et al., 2025).

To construct a semantically coherent latent space, we leverage the observation that states occurring successively along a trajectory share similar desirability (Park et al., 2022a; Wang et al., 2019; Hamedi & Shad, 2022), which is further discussed in Appendix F. This property can be formalized as $P[g(s) = g(s')] \geq 1 - \epsilon, \exists \epsilon > 0, \forall (s, s') \in \mathcal{S}_{\text{adj}}$, where $\mathcal{S}_{\text{adj}}$ denotes the set of adjacent state pairs within trajectories. Consequently, semantic coherence could be promoted by enforcing a constraint that embeddings of such state pairs remain close, as follows:

$$\|\phi(s') - \phi(s)\|_2 \leq \delta_0 \qquad \forall (s, s') \in \mathcal{S}_{\text{adj}}, \quad (5)$$

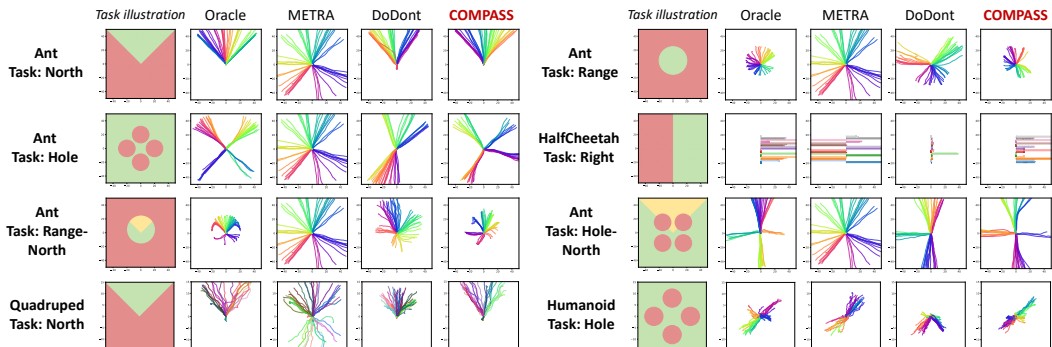

Figure 2: Visualizations of skills learned by COMPASS and baseline methods, by plotting x-y (or x) trajectories sampled from the learned policies. Different colors represent distinct skills $z$. In task illustrations, human-undesirable regions ($y = 0$) are highlighted in red, desirable regions ($y = 2$) in yellow, and neutral regions in green. COMPASS effectively aligns diverse skills with human intent.

where $\delta_0 > 0$ is a constant. As shown in Park et al. (2023b), this local constraint implies a global Lipschitz condition with respect to temporal distance (Kaelbling, 1993; Hartikainen et al., 2019; Durugkar et al., 2021): $\|\phi(s_1) - \phi(s_2)\| \leq \delta_0 d_{\text{temp}}(s_1, s_2), \forall s_1, s_2 \in \mathcal{S}$, where $d_{\text{temp}}(s_1, s_2)$ is the minimum number of steps to transition from $s_1$ to $s_2$. This result connects our semantically coherent DSD approach to Park et al. (2023b).

**Training-free guidance signal construction.** Building upon the semantically coherent latent space, we now construct the guidance signal $w(s)$ in a training-free manner. The key idea is that for any state $s$, its human desirability can be inferred from its distances to labeled states within the semantically coherent latent space. Specifically, given a small set of human-labeled states $\mathcal{D} = \mathcal{D}_0 \cup \mathcal{D}_1 \cup \mathcal{D}_2$, we compute the minimum L2-distance from $s$ to each label set in the latent space:

$$d_\phi(s, \mathcal{D}') = \min_{s_0 \in \mathcal{D}'} \|\phi(s_0) - \phi(s)\|, \quad \mathcal{D}' \in \{\mathcal{D}_0, \mathcal{D}_1, \mathcal{D}_2\}. \tag{6}$$

We then define $w(s)$ as a soft assignment over the three desirability levels:

$$w(s) = \text{softmax}\left([-d_\phi(s, \mathcal{D}_0), -d_\phi(s, \mathcal{D}_1), -d_\phi(s, \mathcal{D}_2)]\right)[0, 1, 2]^\top, \tag{7}$$

where $\text{softmax}([x_1, \ldots, x_n]) = [\exp(x_1)/Z, \ldots, \exp(x_n)/Z]$ and $Z = \sum_{i=1}^n \exp(x_i)$. This formulation intuitively assigns higher values to states closer to "good" regions and lower values to those near "bad" regions, ensuring $w(s)$ accurately reflects relative desirability.

This construction is not merely intuitive but also theoretically grounded. As formally established in Proposition 1 (proof in Appendix C), when the constructed $w(s)$ is interpreted as a classifier, its asymptotic error rate is bounded by twice the Bayes error rate. This result guarantees the reliability of our training-free guidance signal $w(s)$.

**Proposition 1.** *We consider the classifier $\hat{g}(s)$ derived from the guidance signal $w(s)$, where $\hat{g}(s) = \arg\max_k \frac{\exp(-d_\phi(s, \mathcal{D}_k))}{\sum_{j=0}^2 \exp(-d_\phi(s, \mathcal{D}_j))}$. The asymptotic expected error rate of the classifier $\hat{g}(s)$ is bounded by the Bayes error rate $P^*(s)$, as follows:*

$$P(\hat{g}(s) \neq g(s)) \leq 2P^*(s) - \frac{3}{2}[P^*(s)]^2, \tag{8}$$

**Full Objective.** Combining them all, we employ the temporal distance as the distance metric $d(x, y)$ in DSD to maintain semantic coherence of the latent skill space, setting $\delta_0 = 1$ for consistency with Park et al. (2023b). The constructed guidance signal $w(s)$ serves as a weighting factor for the DSD objective, leading to the following objective:

$$\sup_{\pi, \phi} \mathbb{E}_{\tau, z}\left[\sum_{t=0}^{T-1} (\phi(s_{t+1}) - \phi(s_t))^\top z\right] \text{ s.t. } w(s)\|\phi(s') - \phi(s)\|_2 \leq 1, \ \forall(s, s') \in \mathcal{S}_{\text{adj}}, \tag{9}$$

Table 1: Safe state coverage results of COMPASS and baselines. For tasks with additional "good" labels, ① refer to *composite safe coverage*, ② refer to *weighted composite safe coverage*. The orange and gray shading represent the best and oracle performances, respectively. COMPASS achieves superior performance across tasks. Table 6 provides safe state ratio results.

| Method | Ant North | Ant Range | Ant Hole | HalfCheetah Right | Quadruped North | Humanoid Hole |
|---|---|---|---|---|---|---|
| **Oracle** | **1381.40** ±150.14 | **620.80** ±35.52 | **1295.60** ±144.86 | **97.80** ±4.87 | **112.60** ±18.05 | **75.20** ±9.39 |
| DIAYN | -4.20 ±0.45 | 4.20 ±0.45 | 4.20 ±0.45 | 0.00 ±0.00 | -4.20 ±0.84 | 3.60 ±0.55 |
| LSD | -1056.80 ±515.24 | -916.80 ±589.79 | 933.20 ±536.19 | -51.00 ±11.77 | -0.60 ±12.72 | 4.20 ±0.45 |
| METRA | -1425.80 ±756.14 | -1247.40 ±147.97 | **1179.00** ±147.26 | -8.40 ±4.16 | -200.80 ±77.72 | 21.60 ±10.81 |
| DoDont* | 1307.20 ±188.33 | -427.60 ±224.09 | 1132.20 ±171.13 | 82.80 ±9.60 | 115.20 ±41.60 | 75.80 ±14.45 |
| COMPASS | **1333.20** ±129.10 | **362.20** ±94.55 | 1149.20 ±127.05 | **102.20** ±4.32 | **128.40** ±44.60 | **80.60** ±25.01 |

| Method | Ant Range-North ① | Ant Range-North ② | Ant Hole-North ① | Ant Hole-North ② | HalfCheetah Not-Flip | Safety-Gym Hazard |
|---|---|---|---|---|---|---|
| **Oracle** | **501.40** ±45.77 | **842.60** ±44.34 | **1143.00** ±77.27 | **1771.60** ±213.72 | **209.60** ±14.03 | **-20.80** ±7.56 |
| DIAYN | 4.20 ±0.45 | 4.20 ±0.45 | 4.20 ±0.45 | 4.20 ±0.45 | 8.80 ±14.10 | -34.80 ±11.45 |
| LSD | -916.80 ±589.79 | -762.40 ±508.72 | 925.20 ±463.76 | 1244.20 ±651.89 | 71.20 ±15.61 | -42.80 ±12.46 |
| METRA | -1247.40 ±147.97 | -1095.00 ±249.60 | **1219.00** ±133.38 | 1582.00 ±171.94 | 187.20 ±10.59 | -34.80 ±14.80 |
| DoDont* | -290.20 ±241.79 | -10.00 ±259.25 | 1135.40 ±294.13 | 1566.60 ±414.17 | 195.00 ±13.55 | -37.60 ±13.22 |
| COMPASS | **380.40** ±145.54 | **622.00** ±199.28 | 1122.00 ±190.43 | **1769.80** ±218.95 | **215.60** ±3.05 | **-16.00** ±10.68 |

However, directly optimizing Eq. 9 is challenging, as the guidance signal $w(s)$ is embedded in the latent space constraint, and directly impacts the update of the latent space $\phi(s)$. This coupling leads to instability, especially since our $w(s)$ is updated dynamically with incoming human feedback. To address this, we follow Kim et al. (2024) and derive an equivalent but more practical objective of Eq. 9:

$$\sup_{\pi,\phi} \mathbb{E}_{\tau,z}\left[\sum_{t=0}^{T-1} w(s_t)\left(\phi(s_{t+1}) - \phi(s_t)\right)^\top z\right] \text{ s.t. } \|\phi(s') - \phi(s)\|_2 \leq 1, \ \forall (s,s') \in \mathcal{S}_{\text{adj}}, \quad (10)$$

Specifically, this derivation is based on a variable substitution, where we replace the latent function with a scaled version, $\phi'(s) = \phi(s)/w(s)$. Appendix D provides the formal derivation. This reformulation offers a crucial advantage: it decouples the guidance signal $w(s)$ from the DSD's latent space learning. It preserves the stability and latent space structure of the original DSD framework, while injecting human guidance by simply scaling DSD's intrinsic reward $r(s,z,s') = w(s)(\phi(s') - \phi(s))^\top z$.

Eq. 10 could be optimized by updating the latent $\phi$ and the Lagrange multiplier $\lambda$ to maximize Eq. 11 and Eq. 12, and updating the policy $\pi$ to maximize the accumulated intrinsic reward in Eq. 13:

$$\mathcal{J}^\phi = \mathbb{E}_{(s,z,s')\sim\mathcal{D}}[w(s)(\phi(s') - \phi(s))^\top z + \lambda \min(\epsilon, 1 - \|\phi(s') - \phi(s)\|_2^2)] \quad (11)$$

$$\mathcal{J}^\lambda = \mathbb{E}_{(s,z,s')\sim\mathcal{D}}[\lambda \min(\epsilon, 1 - \|\phi(s') - \phi(s)\|_2^2)] \quad (12)$$

$$r(s,z,s') = w(s)(\phi(s') - \phi(s))^\top z \quad (13)$$

### 3.3 ACTIVE QUERY SELECTION

As described in Section 3.2, sparse human labels are used to construct the training-free guidance signal $w(s)$. A natural question arises: *How to efficiently collect informative human feedback?* Query selection (Lee et al., 2021; Hu et al., 2023) addresses this by identifying the most valuable segments for evaluation, thereby maximizing the utility of each label. This is particularly beneficial in our GSD framework, where human feedback is sparse and costly.

The effectiveness of our training-free guidance signal, $w(s)$, relies on accurately estimating human desirability for any state via its nearest labeled neighbors in the latent space. This requires the labeled states set $\mathcal{D}$ to sufficiently cover the state space. To achieve this, we propose a query selection mechanism that prioritizes less-visited states by maximizing their state entropy in the labeled states $H_s(s) = -\log \Pr_{s\sim\mathcal{D}}(s)$. Since directly computing state entropy is intractable, we use an efficient particle-based entropy estimation (Singh et al., 2003; Liu & Abbeel, 2021b):

$$H_s(s) \approx \log\left(1 + \frac{1}{k}\sum_k \|s - s^k\|\right), \quad (14)$$

Table 2: Zero-shot downstream task performance on Ant tasks. We report the average and the best performance of the learned skills.

| Method | Hole (avg) | North (avg) | Range (avg) | Hole (best) | North (best) | Range (best) |
|---|---|---|---|---|---|---|
| **Oracle** | **938.23** $\pm$ 201.76 | **1111.08** $\pm$ 257.30 | **-595.66** $\pm$ 801.53 | **1165.42** $\pm$ 89.36 | **1521.60** $\pm$ 91.41 | **717.70** $\pm$ 28.89 |
| DIAYN | 209.85 $\pm$ 5.78 | -2833.11 $\pm$ 1539.28 | 210.09 $\pm$ 5.86 | 243.62 $\pm$ 17.53 | 215.07 $\pm$ 1.37 | 244.73 $\pm$ 19.49 |
| LSD | -34.05 $\pm$ 777.39 | -2017.02 $\pm$ 1493.83 | -894.28 $\pm$ 455.99 | 1112.01 $\pm$ 386.15 | 1054.53 $\pm$ 467.55 | 474.65 $\pm$ 180.11 |
| METRA | 224.07 $\pm$ 555.11 | -1897.47 $\pm$ 1431.75 | -1149.32 $\pm$ 275.56 | 1193.03 $\pm$ 30.62 | 1249.26 $\pm$ 104.01 | 265.29 $\pm$ 474.71 |
| DoDont* | 333.40 $\pm$ 633.35 | 277.15 $\pm$ 1519.21 | -1045.95 $\pm$ 374.46 | 1155.58 $\pm$ 48.08 | 1397.12 $\pm$ 45.58 | 540.25 $\pm$ 62.46 |
| COMPASS | 650.67 $\pm$ 637.03 | 1054.25 $\pm$ 376.79 | -913.48 $\pm$ 715.64 | 1251.92 $\pm$ 72.67 | 1460.44 $\pm$ 53.12 | 673.55 $\pm$ 65.86 |

where $s^k$ denotes the $k$-th nearest neighbors of state $s$ in the labeled state dataset $\mathcal{D}$. Based on this, we design the query selection score $I(\sigma)$ for segment $\sigma$ as $I(\sigma) = \sum_{s \in \sigma} H_s(s)$. Details on the query selection mechanism are provided in Appendix B.

### 3.4 IMPLEMENTATION DETAILS

**Algorithm outline.** Algorithm 1 and Fig. 1 outline the procedure of COMPASS. Building on the DSD framework, we iteratively collect feedback from the learned skills (lines 5∼10) and immediately employ the feedback to construct the guidance signal (line 11), enabling the guidance signal to be updated online with the skills.

**Computational efficiency.** Although the computation of the guidance signal $w(s)$ involves calculating distances $d_\phi(s, D)$ over all labeled states, the computational burden is not heavy. This is because the number of labeled states is small, and we cache the embeddings $\phi$ of these states to further accelerate the process.

**Guidance signal smoothing.** The training-free construction of $w(s)$ enables efficient guidance, but suffers from abrupt changes after each feedback session. This particularly impacts early-stage skill learning, when both the latent space and policy are underdeveloped. To mitigate this, we employ a smoothing mechanism: $w_e(s) = (1 - \beta_e) \cdot w(s) + \beta_e \cdot 1$, $\beta_e = \max(0, 1 - k_\beta \cdot \frac{e}{T^e})$, where $k_\beta$ is a hyperparameter controlling the decay of $\beta_e$, $e$ is the skill learning epoch index ($e = 1, 2, \ldots, T^e$), and $T^e$ is the total epoch number. During experiments, we use $w_e(s)$ as the guidance signal in epoch $e$. This introduces a smooth transition from pure exploration to guided exploration, with a smaller $k_\beta$ resulting in a slower transition; At $k_\beta = \infty$, the algorithm remains pure GSD, while $k_\beta = 0$ corresponds to pure USD. By gradually introducing the guidance signal, this approach improves the stability of skill learning, as validated in the ablation studies in Section 4.4.

## 4 EXPERIMENT

We conduct extensive experiments to answer the following questions: *Q1*: Can a training-free guidance signal, constructed from minimal human labels, reliably promote diverse behaviors toward safe and meaningful regions? *Q2*: Is COMPASS effective in complex pixel-based settings, where state representations must be inferred from raw pixels? *Q3*: What is the contribution of each proposed technique in COMPASS?

### 4.1 SETUP

**Domains and guidance tasks.** We evaluate COMPASS on five complex robotic locomotion environments: state-based Ant and HalfCheetah from OpenAI Gym (Todorov et al., 2012; Brockman et al., 2016), pixel-based Quadruped and Humanoid from DMControl (Tassa et al., 2018), and state-based Safety Gym (Ray et al., 2019). To assess COMPASS's alignment with human intent, we design tasks with varying guidance types: **(1) Direction**, where the agent moves towards a specific direction (e.g., *North* and *Right*). **(2) Range**, where the agent explores within a range (*Range*). **(3) Hazard avoidance**, where the agent avoids hazardous areas (*Hole* and *Hazard*). **(4) Unsafe behaviour avoidance**, where the agent avoids unsafe actions (*Not-Flip*). **(5) Composite tasks**, which combine multiple guidance types, requiring hazard avoidance while encouraging directional movement (*Range-North* and *Hole-North*). These tasks are illustrated in Figure 2, with further details in Appendix H.1.

**Feedback collection.** Following prior works (Lee et al., 2021), we use an oracle teacher for systematic evaluation. The teacher provides feedback based on human-defined task rules, aligning with human intent. A segment is labeled as "bad" if it contains any undesirable state, "good" if all states are desirable, and "neutral" otherwise. This conservative labeling reflects human preferences for safety, disfavoring even brief entries into hazardous regions. Appendix H.3 provides further details.

**Baselines and implementation.** We compare COMPASS with three groups of baselines: **(1) USD methods**, including a mutual information-based method, DIAYN (Eysenbach et al., 2019), and two DSD methods, LSD (Park et al., 2022b) and METRA (Park et al., 2023b). **(2) GSD method**, specifically an online variant of DoDont (Kim et al., 2024) (DoDont*). Since DoDont requires training an instruction network with pre-collected expert data, which is unavailable in our setting, we use online-collected data similar to COMPASS. **(3) an Oracle version of COMPASS** (Oracle), which employs a manually designed $w(s)$ in COMPASS to provide ideal guidance signals, serving as the performance upper bound. For constructing the guidance signal, COMPASS uses 40 labeled segments of length $H = 20$ for most tasks. Appendix H.3 provides further details.

**Metrics.** We employ three main metrics for evaluation: **(1) Safe state coverage**, which measures the agent's ability to explore the state space while avoiding hazardous regions. Following (Kim et al., 2024), this metric assigns a value $+1, -1$ to safe and unsafe areas, and computes state coverage by counting the unique $1 \times 1$ x-y bins (or 1-unit x-axis bins for HalfCheetah tasks) visited by the agent. **(2) Safe state ratio**, which quantifies the proportion of visited safe bins among all visited bins, serves as a normalized safe state coverage. **(3) Downstream task performance**, which evaluates the utility of learned skills in downstream tasks. For all metrics, we report the average and standard deviation across 5 random seeds. Appendix H.3 provides more details.

## 4.2 Main Results

**Hazard area avoidance with sparse feedback.** We first assess COMPASS's ability to avoid static hazardous areas in state-based Ant and HalfCheetah environments. As shown in Fig. 2, COMPASS successfully learns skills constrained to safe regions in Ant North, Range, Hole, and HalfCheetah Right tasks, while unsupervised baselines explore indiscriminately, often visiting hazardous areas. Tables 1 and 6 quantitatively confirm this, with COMPASS achieving near-oracle performance in safe state coverage and safe state ratio, surpassing all baselines on most tasks. These results demonstrate that our training-free guidance signal provides a robust and effective safety constraint. In contrast, DoDont, which depends on a trained instruction network, performs worse, likely due to the network's instability with limited feedback data.

**Enhanced exploration efficiency with additional "good" labels.** Beyond merely avoiding bad regions, we evaluate COMPASS in tasks that incorporate both positive ("good") and negative ("bad") feedback labels, specifically in Ant Range-North and Hole-North tasks. This allows for assessing COMPASS's ability to not only avoid hazards but also actively promote exploration toward human-preferred regions. To quantify this, we extend the safe state coverage metric to *composite safe coverage* and *weighted composite safe coverage*, which assign $+1, +1, -1$ and $+2, +1, -1$ to good, neutral, and bad regions, respectively. As shown in Tables 1, COMPASS surpasses almost all baselines in these tasks. Visualizations in Fig. 2 further show dense and uniform skill trajectory coverage within "good" regions, indicating COMPASS's flexibility as a unified framework for encouraging desired behaviors and while avoiding undesirable ones.

**Unsafe behavior avoidance.** Beyond avoiding static hazardous areas, we further assess COMPASS's capability to prevent dynamic unsafe behaviors. In the HalfCheetah Not-Flip task, the agent must learn diverse locomotion skills while avoiding potentially damaging flipping behaviors. As shown in Table 1, COMPASS achieves the highest safe state coverage. This result shows that COMPASS can effectively avoid not just static hazards but also dynamic undesirable behaviors.

**Effectiveness on pixel-based tasks.** To assess COMPASS's ability in complex environments, we evaluate COMPASS on pixel-based Quadruped and Humanoid environments, each with 100 feedback. As shown in Fig. 2, COMPASS successfully avoids hazards in both the Quadruped North and Humanoid Hole tasks, despite the high-dimensional states. Quantitative results in Tables 1 and 6 align with the visualizations. These results show that the COMPASS's training-free guidance mechanism effectively leverages the coherent latent space, generalizing beyond state-based inputs.

Table 3: Comparison of safe state coverage of COMPASS and various query selection methods. Appendix E provides more results.

| Method | Ant Hole | Ant North |
|---|---|---|
| COMPASS | $1149.20 \pm 127.05$ | $1333.20 \pm 129.10$ |
| Uniform | $633.80 \pm 279.41$ | $1257.80 \pm 189.01$ |
| Uncertainty | $1034.40 \pm 468.49$ | $1059.20 \pm 79.16$ |

Table 4: Safe state coverage results of COMPASS using different smooth speed $k_\beta$. Appendix E provides more results.

| $k_\beta$ | Ant Hole | Ant North |
|---|---|---|
| 3 | $1041.60 \pm 226.01$ | $1349.00 \pm 162.01$ |
| 5 | $1149.20 \pm 127.05$ | $1333.20 \pm 129.10$ |
| 10 | $1068.40 \pm 321.47$ | $1251.80 \pm 99.38$ |
| $\infty$ | $991.60 \pm 188.15$ | $1120.80 \pm 420.01$ |

## 4.3 DOWNSTREAM TASK PERFORMANCE

We evaluate the utility of skills learned by COMPASS on downstream tasks, both in zero-shot settings and after task-specific hierarchical control. We design two types of tasks: an Ant motion task with safety penalties for entering hazardous regions, and a HalfCheetah goal-reaching task that penalizes unsafe behaviors such as flipping, with details provided in Appendix H.2.

In the Ant task, we evaluate all skills in a zero-shot manner. As shown in Table 2, COMPASS achieves the highest average and best performance across all task variants, demonstrating that the learned skills effectively avoid undesirable states while retaining high mobility and task relevance. For the HalfCheetah task, we train a high-level

Table 5: Downstream task performance on the HalfCheetah task. We train a hierarchical controller to select low-level frozen skills, and report the average and the best performance.

| Method | Avg | Best |
|---|---|---|
| DIAYN | $-6.12 \pm 22.73$ | $159.20 \pm 89.00$ |
| LSD | $-17.33 \pm 29.59$ | $155.20 \pm 87.19$ |
| METRA | $-15.78 \pm 30.40$ | $177.20 \pm 32.81$ |
| DoDont* | $-3.32 \pm 24.14$ | $159.20 \pm 89.00$ |
| COMPASS | $\mathbf{4.03} \pm 4.73$ | $\mathbf{199.00} \pm 0.00$ |

controller to select from the frozen skill set, detailed in H.3. Table 5 shows that COMPASS outperforms all baselines, confirming that the learned skills are highly effective for hierarchical downstream task solving. These results demonstrate that COMPASS acquires semantically meaningful and useful skills, enabling strong downstream performance.

## 4.4 ABLATION STUDY

**Analysis on query selection methods.** To evaluate the effectiveness of COMPASS's query selection method, we compare it with two baselines: a uniform sampling method (Uniform), and an uncertainty-focused method (Uncertainty) that prioritizes states with large guidance signal entropy:

$$H_{\mathrm{w}}(s) = \mathbb{H}[\mathrm{softmax}\left([-d_\phi(s, \mathcal{D}_0), \; -d_\phi(s, \mathcal{D}_1), \; -d_\phi(s, \mathcal{D}_2)]\right)]. \tag{15}$$

As shown in Table 3, COMPASS consistently outperforms both baselines, achieving higher safe state coverage and safe ratio across all tasks. While the Uncertainty approach refines preference boundaries, it often neglects exploration and fails to cover diverse regions of the state space. In contrast, COMPASS prioritizes under-explored regions, leading to a more effective guidance signal. These results underscore the critical role of exploration in query selection of GSD.

**Robustness to hyperparameters.** To examine the impact of the smooth parameter $k_\beta$, which controls the transition speed from USD to GSD, we conduct an ablation study. Table 4 shows that incorporating $k_\beta$ improves performance, but overly slow transitions weaken the guidance signals, impairing skill learning. Based on these results, we set $k_\beta = 5$ in the main experiments, as it consistently outperforms the configuration without smoothing.

## 5 RELATED WORK

**Unsupervised skill discovery (USD).** Unsupervised skill discovery (USD) aims to learn a set of distinguishable policies that collectively cover the state space using unlabeled data, without task-specific rewards, to facilitate downstream tasks. A common USD objective is maximizing the mutual information (MI) $I(s, z)$ between states $s$ and latent skills $z$ (Eysenbach et al., 2019; Sharma et al., 2020; Liu & Abbeel, 2021a; Laskin et al., 2022). Though it can yield diverse behaviors, maximizing MI often fails to promote broad exploration, leading to static behaviors (Park et al., 2022b; 2023b).

To address this, Distance-maximizing Skill Discovery (DSD) methods have been introduced, which link latent-space distances to state-space distances to promote coverage. METRA (Park et al., 2023b) formally derives the DSD objective by replacing MI with the Wasserstein dependency measure. DSD allows any distance function $d(\cdot, \cdot) : \mathcal{S} \times \mathcal{S} \to \mathbb{R}_0^+$ to encourage exploration of different state sub-spaces. Examples include Euclidean distance to encourage geometrically longer travel (Park et al., 2022b), negative log-likelihoods of an estimated transition probability to prioritize rarely visited states (Park et al., 2023a), and temporal distance to encourage temporally distant exploration (Park et al., 2023b). However, these methods often lead to uniform exploration within sub-spaces, which may result in unnecessary or even unsafe exploration.

**Guided skill discovery (GSD).** Recent works in GSD incorporate prior knowledge to reduce unnecessary exploration in USD, leveraging expert trajectories (Klemsdal et al., 2021; Kim et al., 2024) or analytical constraint formulas (Kim et al., 2023). Specifically, Klemsdal et al. (2021) and DoDont (Kim et al., 2024) train classifiers to distinguish expert trajectories from others. Klemsdal et al. (2021) further uses the classifier's encoder as a state projection to encourage exploring the expert-concerned state subspace, while DoDont uses the classifier's probability output as the distance function in DSD. Kim et al. (2023) employs Lagrangian Q-learning to ensure skill safety. Recent studies explore utilizing pairwise human preferences (Hussonnois et al., 2023; 2025) to learn a human-aligned reward model. These models then guide the skill discovery process by identifying preferred regions (Hussonnois et al., 2023) or encouraging alignment between skills and human values (Hussonnois et al., 2025). Despite these advancements, deriving expert trajectories or constraint formulas is often challenging and impractical in complex tasks, and classifiers or reward models trained on limited data can be unstable. This paper aims to address these limitations.

## 6 CONCLUSION

In this paper, we propose COMPASS, a training-free guided skill discovery framework that effectively aligns exploration with human intent using sparse feedback. By enforcing a semantically coherent skill latent space, COMPASS constructs a dense guidance signal from minimal human feedback, and integrates this guidance signal into DSD objectives, eliminating the need for expert demonstrations or auxiliary model training. COMPASS further employs an active query strategy to ensure the guidance signal's accuracy. Experiments show that COMPASS learns diverse, human-preferred skills, avoids unsafe behaviors, and facilitates downstream tasks. Our work presents a simple yet powerful approach to integrating human guidance into unsupervised skill discovery.

## REPRODUCIBILITY STATEMENT

To facilitate reproducibility, we provide anonymous code in Appendix H.3. and detailed experimental configurations. Hyperparameters, network architectures, and training procedures are fully documented in Appendix H. The guidance signal construction and active query selection mechanism are described algorithmically in Section 3 and Appendix B. All results are reported over multiple seeds, and metrics are defined explicitly in Section 4.1.

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

## A  LLM USAGE STATEMENT

We utilized large language models (LLMs) to enhance the writing quality of this paper. All content was initially written by the authors and then processed through the LLM to correct grammar, improve word choice, and refine expressions. The authors have reviewed every sentence to ensure accuracy.

## B  ALGORITHM

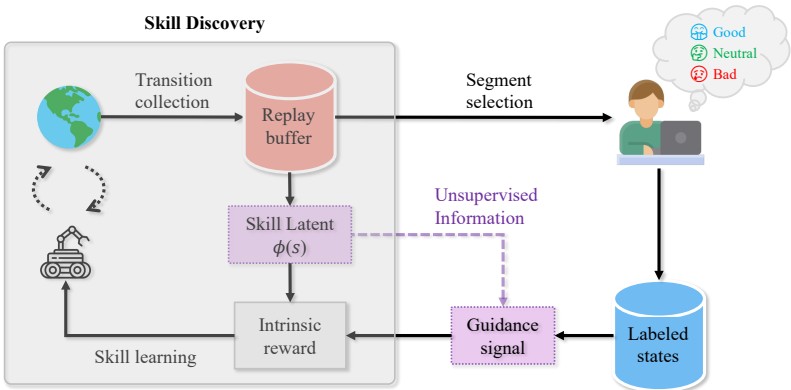

Figure 3: An overview of the proposed COMPASS. COMPASS performs skill discovery and guidance signal learning simultaneously. The guidance signal is constructed by leveraging both rich unsupervised information from the learned skill latent space and human feedback on labeled states, without relying on expert-level data.

We illustrate the full process of COMPASS in Algorithm 2 and Fig. 3, and illustrate the query selection strategy in detail in Algorithm 3.

---

**Algorithm 2** COMPASS (more detailed)

---

**Require:** Feedback frequency $K$, total feedback number $N_{\text{total}}$, number of queries per feedback session $M$, total epoch number $T^{\text{e}}$

1: Initialize replay buffer $\mathcal{B}$, feedback buffer $\mathcal{D}_0$, $\mathcal{D}_1$, $\mathcal{D}_2$
2: **for** each epoch $e = 1, 2 \ldots, T^{\text{e}}$ **do**
3:     Sample skill $z \sim p(z)$
4:     Rollout with policy $\pi(a|s, z)$ and store $(s, a, s')$ into $\mathcal{B}$
5:     **if** epoch % $K = 0$ and $|\mathcal{D}_0| + |\mathcal{D}_1| + |\mathcal{D}_2| < N_{\text{total}}$ **then**
6:         Select segments $\{\sigma_i\}_{i=1}^{M} \sim \mathcal{B}$ using the query selection method in Section 3.3 and Algorithm 3
7:         Query labelers for feedback $\{y_i\}_{i=1}^{M}$, $y_i \in \{0, 1, 2\}$
8:         Save labeled states into feedback buffer, $\mathcal{D}_0 \leftarrow \mathcal{D}_0 \cup \{s : s \in \sigma_i, y_i = 0\}_{i=1}^{M}$ (similar for $\mathcal{D}_1$ and $\mathcal{D}_2$)
9:     **end if**
    // *Update the policy*
10:     Sample transitions from $\mathcal{B}$, and calculate the guidance signal $w(s)$ with Eq. 7 and the smooth mechanism in Section 3.4
11:     Update the skill latent $\phi(s)$ with Eq. 11
12:     Update the Lagrange multipler $\lambda$ with Eq. 12
13:     Update the skill conditioned policy $\pi(a|s, z)$ with Eq. 13
14: **end for**

---

---

**Algorithm 3** QUERY SELECTION

---

**Require:** Number of candidate queries $N_c$, number of queries per feedback session $M$
1: Sample $N_c$ segments $\{\sigma_i\}_{i=1}^{N_c}$
2: Initialize query selection vector of shape $N_c$ with zeros: $\hat{I} = [0, 0, \ldots, 0]$.
3: **for** each segment $\sigma_i$ **do**
4:     Calculate selection score and store it in $\hat{I}$: $\hat{I}_i \leftarrow I(\sigma_i)$
5: **end for**
6: Select $M$ queries with the top-$M$ query selection score $\hat{I}$

---

This online update differentiates our method from existing GSD approaches by using guidance derived from expert trajectories or analytical constraints. The online update allows the guidance signal to evolve jointly with the skills, thereby leveraging the unsupervised skill latent space and the collected trajectories. Specifically, the unsupervised skill latent space enables training-free construction of the guidance signal. Additionally, trajectories collected alongside the skills enable interactive query collection, eliminating the need for expert trajectories to train the guidance model.

## C  PROOF

In this section, we theoretically analyze the performance of the proposed guidance signal $w(s)$. Since $w(s)$ is derived as an expectation in Eq. 7, we evaluate it by analyzing the estimated human desirability distribution, given as:

$$\text{softmax}\left([-d_\phi(s, \mathcal{D}_0),\ -d_\phi(s, \mathcal{D}_1),\ -d_\phi(s, \mathcal{D}_2)]\right). \tag{16}$$

Denote the labeled dataset as $\mathcal{D} = \{(s_i, y_i)\}_{i=1}^n$, where $y_i \in \{0, 1, 2\}$ represents the human desirability label. For each class $k$, let $\mathcal{D}_k = \{(s_i, y_i) \in \mathcal{D} \mid y_i = k\}$. Let the true human desirability function $g(s) : \mathcal{S} \to \{0, 1, 2\}$ be a random variable, with conditional probabilities $\eta_k(s) = P(g(s) = k \mid s)$ for $k \in \{0, 1, 2\}$. We consider a classifier $\hat{g}(s)$ induced from $w(s)$. For any given $s$, $\hat{g}(s)$ calculates the distances $d_{\phi,k}(s) = \min_{(s_i, y_i) \in \mathcal{D}_k} \|\phi(s_i) - \phi(s)\|$, estimates the probabilities as $\hat{\eta}_k(s) = \frac{\exp(-d_{\phi,k}(s))}{\sum_{j=0}^2 \exp(-d_{\phi,j}(s))}$, and assigns the state $s$ to the class $\hat{g}(s) = \arg\max_k \hat{\eta}_k(s)$.

In addition, we make the following assumptions:

- The labeled dataset $\mathcal{D} = \{(s_i, y_i)\}_{i=1}^n$ is sampled independently and identically distributed (i.i.d.).

- The number of samples in each class, $n_k = |\mathcal{D}_k|$, satisfies $n_k \to \infty$ as $n \to \infty$.

**Lemma 1** (Cover & Hart (1967)). *Consider a K-class classification problem where the input set is $\mathcal{S}$, and the ground labels are specified by a random variable $g(s) : \mathcal{S} \to \{1, 2, \ldots K\}$ with conditional class probabilities $\eta_k(s) = P(g(s) = k \mid s)$. Assume the labeled data $\mathcal{D} = \{(s_i, y_i)\}_{i=1}^n$ is i.i.d. sampled. For the nearest neighbor classifier $\hat{g}_{NN}(s)$ that assigns to a test point $s$ the label of its closest labeled point $\hat{g}_{NN}(s) = y^*$, $s^* = \arg\min_{s_i \in \mathcal{D}} |\phi(s_i) - \phi(s)|$, the asymptotic error rate satisfies:*

$$\lim_{n \to \infty} P(\hat{g}_{NN}(s) \neq g(s)) = \mathbb{E}\left[1 - \sum_{k=1}^K \eta_k(s)\eta_k(s^*)\right] \leq 2P^*(s) - \frac{K}{K-1}[P^*(s)]^2, \tag{17}$$

*where $P^*(s) = 1 - \max_k \eta_k(s)$ is the Bayes error rate.*

*Proof.* The classification error occurs when $y^* \neq g(s)$. Given $s$ and its nearest neighbor $s^*$, the error probability is:

$$P(y^* \neq g(s) \mid s, s^*) = 1 - P(y^* = g(s) \mid s, s^*). \tag{18}$$

Conditional on $s$, the distribution of $g(s)$ is determined by $\eta_k(s)$; conditional on $s^*$, the distribution of $y^*$ is determined by $\eta_k(s^*)$ (since $y^*$ is a realization of $g(s^*)$). As $y^*$ and $g(s)$ are conditionally independent given $s$ and $s^*$, we have

$$P(y^* = g(s) \mid s, s^*) = \sum_{k=1}^K P(g(s) = k \mid s)P(y^* = k \mid s^*) = \sum_{k=1}^K \eta_k(s)\eta_k(s^*), \tag{19}$$

Taking expectation over $s$ and $s^*$, we have

$$P(\hat{g}_{\text{NN}}(s) \neq g(s)) = \mathbb{E}\left[1 - \sum_{k=1}^K \eta_k(s)\eta_k(s^*)\right]. \tag{20}$$

As $n \to \infty$, $s^*$ converges to $s$ (due to denseness of the point set), and if $\eta_k$ is continuous, then $\eta_k(s^*) \to \eta_k(s)$. Therefore, the asymptotic error rate becomes

$$\lim_{n \to \infty} P(\hat{g}_{\text{NN}}(s) \neq g(s)) = \mathbb{E}\left[1 - \sum_{k=1}^K \eta_k(s)^2\right]. \tag{21}$$

To bound this expression, let $\eta_{(1)} \geq \eta_{(2)} \geq \cdots \geq \eta_{(K)}$ be the ordered values of $\eta_k(s)$, so $\max_k \eta_k(s) = \eta_{(1)}$ and $P^*(s) = 1 - \eta_{(1)}$. The sum of squares $\sum \eta_k(s)^2$ is minimized when the remaining probability $1 - \eta_{(1)}$ is distributed uniformly among the other $K - 1$ classes. Therefore, we have

$$\sum_{k=1}^K \eta_k(s)^2 \geq \eta_{(1)}^2 + \frac{(1 - \eta_{(1)})^2}{K-1}. \tag{22}$$

Using $P^*(s) = 1 - \eta_{(1)}$, we have

$$1 - \sum \eta_k(s)^2 \leq P^*(s)(2 - P^*(s)) - \frac{(P^*(s))^2}{K-1} = 2P^*(s) - \frac{K}{K-1}[P^*(s)]^2. \quad (23)$$

Substituting it into Eq. 21 concludes the proof. $\qquad\square$

**Proposition 1.** *We consider the classifier $\hat{g}(s)$ derived from the guidance signal $w(s)$, where $\hat{g}(s) = \arg\max_k \frac{\exp(-d_\phi(s, \mathcal{D}_k))}{\sum_{j=0}^2 \exp(-d_\phi(s, \mathcal{D}_j))}$. The asymptotic expected error rate of the classifier $\hat{g}(s)$ is bounded by the Bayes error rate $P^*(s)$, as follows:*

$$P(\hat{g}(s) \neq g(s)) \leq 2P^*(s) - \frac{3}{2}[P^*(s)]^2, \quad (8)$$

*Proof.* Consider the nearest neighbor classifier $\hat{g}_{\text{NN}}(s)$, which selects the label $y^*$ of the labeled point $s^*$ closest to $s$:

$$s^* = \arg\min_{s_i \in \mathcal{D}} \|\phi(s_i) - \phi(s)\|, \quad \hat{g}_{\text{NN}}(s) = y^*. \quad (24)$$

For any $(s^*, y^*)$ selected by $\hat{g}_{\text{NN}}(s)$, we have $d_{\phi, y^*}(s^*) < d_{\phi, y_i}(s_i), \forall(s_i, y_i) \in \mathcal{D}$. Then, $\hat{\eta}_{y^*}(s) \geq \hat{\eta}_k(s)$, $\forall k \in \{0, 1, 2\}$. Therefore, $\hat{g}(s)$ always has the same estimation as the nearest neighbor classifier $\hat{g}_{\text{NN}}(s)$. The classifier $\hat{g}(s)$ is equvalent to a nearest neighbor classifier $\hat{g}_{\text{NN}}(s)$.

According to Lemma 1, for this 3-class classification problem, as $n \to \infty$, the error rate of the nearest neighbor classifier satisfies:

$$\lim_{n \to \infty} P(\hat{g}_{\text{NN}}(s) \neq g(s)) \leq 2P^*(s) - \frac{3}{2}[P^*(s)]^2. \quad (25)$$

Therefore, $\hat{g}(s)$ satisfies

$$P(\hat{g}(s) \neq g(s)) \leq 2P^*(s) - \frac{3}{2}[P^*(s)]^2, \quad (26)$$

which concludes the proof. $\qquad\square$

# D  DEVIATION OF THE OBJECTIVE FUNCTION

In this section, we derive the objective function of COMPASS, Eq. 10, from the DSD-form objective function, Eq. 9, in a similar manner as Kim et al. (2024). We assume the guidance signal $w(s)$ is continuous.

We start by restating Eq. 9:

$$\sup_{\pi,\phi} \mathbb{E}_{\tau,z} \left[ \sum_{t=0}^{T-1} \left( \phi(s_{t+1}) - \phi(s_t) \right)^\top z \right] \text{ s.t. } \|\phi(s') - \phi(s)\|_2 \leq w(s)d(s,s'), \ \forall(s,s') \in S_{\text{adj}}. \quad (27)$$

Define a scaled latent function $\phi'(s) \triangleq \frac{\phi(s)}{w(s)}$. As $w(s) \geq 0$ in COMPASS, we derive the following formula approximately.

$$\sup_{\pi,\phi} \mathbb{E}_{\tau,z} \left[ \sum_{t=0}^{T-1} \left( \phi(s_{t+1}) - \phi(s_t) \right)^\top z \right] \text{ s.t. } \|\phi'(s') - \phi'(s)\|_2 \leq d(s,s'), \ \forall(s,s') \in S_{\text{adj}}. \quad (28)$$

The approximation $\frac{\phi(s')}{w(s)} \approx \frac{\phi(s')}{w(s')}$ leverages the continuity of the guidance signal $w(s)$, as $s$ and $s'$ are adjacent states in this context.

Then, we replace $\phi(s)$ with $w(s)\phi'(s)$, deriving

$$\sup_{\pi,\phi} \mathbb{E}_{\tau,z} \left[ \sum_{t=0}^{T-1} w(s_t) \left( \phi'(s_{t+1}) - \phi'(s_t) \right)^\top z \right] \text{ s.t. } \|\phi'(s') - \phi'(s)\|_2 \leq d(s,s'), \ \forall(s,s') \in S_{\text{adj}}.$$

$$(29)$$

which is exactly Eq. 10.

# E    MORE EXPERIMENTAL RESULTS

## E.1    SAFE STATE RATIO RESULTS FOR SECTION 4.2

We report the safe state ratio results in Table 6. For tasks with additional good labels (Ant Range-North and Ant Hole-North), the safe ratio is calculated as the proportion of good and neutral state bins relative to all visited state bins.

As shown in the table, COMPASS outperforms other baselines in 4 out of 6 tasks. Note that DIAYN achieves 100% safe state ratio in Ant Hole and Ant Hole-North tasks. This is primarily because the range of states covered by DIAYN is highly restricted, causing it to visit only good or neutral states. In contrast, COMPASS achieves a superior safe coverage (in Table 1) and safe state ratio simultaneously, demonstrating its effectiveness.

Table 6: Safe state ratio results (%) of COMPASS and baselines. The orange and gray shading represent the best and oracle performances, respectively. COMPASS achieves superior performance across tasks.

| Method | Ant North | Ant Range | Ant Hole | HalfCheetah Right | Quadruped North | Humanoid Hole |
|---|---|---|---|---|---|---|
| **Oracle** | **92.60** ± 1.60 | **94.30** ± 2.00 | **98.90** ± 0.70 | **99.00** ± 0.00 | **79.80** ± 2.90 | **91.20** ± 5.30 |
| DIAYN | 0.00 ± 0.00 | 0.00 ± 0.00 | 100.00 ± 0.00 | 50.00 ± 0.00 | 6.20 ± 8.50 | **100.00** ± 0.00 |
| LSD | 18.30 ± 10.90 | 36.40 ± 26.10 | 69.00 ± 13.50 | 28.50 ± 4.20 | 20.60 ± 24.20 | **100.00** ± 0.00 |
| METRA | 20.40 ± 14.70 | 23.90 ± 2.10 | 74.70 ± 2.20 | 48.00 ± 1.00 | 21.10 ± 10.60 | 82.30 ± 15.40 |
| DoDont* | 93.40 ± 4.70 | 36.70 ± 5.70 | 83.20 ± 4.50 | 86.70 ± 5.30 | 76.00 ± 8.00 | 84.20 ± 4.30 |
| COMPASS | **96.90** ± 2.50 | **80.90** ± 7.80 | 90.60 ± 2.40 | **98.70** ± 0.50 | **87.60** ± 7.50 | 90.50 ± 9.80 |

| Method | Ant Range-North | Ant Hole-North | HalfCheetah Not-Flip | Safety-Gym Hazard |
|---|---|---|---|---|
| **Oracle** | **95.80** ± 1.50 | **98.60** ± 0.30 | **100.00** ± 0.00 | **37.00** ± 4.70 |
| DIAYN | 0.00 ± 0.00 | 100.00 ± 0.00 | 75.20 ± 14.00 | 28.30 ± 7.20 |
| LSD | 36.40 ± 26.10 | 74.50 ± 1.50 | 76.10 ± 5.80 | 23.20 ± 7.80 |
| METRA | 23.90 ± 2.10 | 75.60 ± 2.50 | 89.20 ± 3.20 | 28.30 ± 9.30 |
| DoDont* | 41.30 ± 6.20 | 83.90 ± 5.00 | 91.70 ± 2.80 | 26.50 ± 8.30 |
| COMPASS | **81.40** ± 8.50 | **93.10** ± 4.80 | **100.00** ± 0.00 | **40.00** ± 6.70 |

## E.2    COMPLETE ABLATION RESULTS FOR SECTION 4.4

We provide more ablation results in Table 7 and 8.

Table 7: Comparison of safe state coverage of COMPASS and various query selection methods. Table 9 provides safe state ratio results.

| Method | Ant Hole | Ant North | Ant Range |
|---|---|---|---|
| COMPASS | 1149.20 ± 127.05 | 1333.20 ± 129.10 | 329.80 ± 110.41 |
| Uniform | 633.80 ± 279.41 | 1257.80 ± 189.01 | 40.80 ± 130.20 |
| Uncertainty | 1034.40 ± 468.49 | 1059.20 ± 79.16 | -79.60 ± 302.86 |

Table 8: Safe state coverage results of COMPASS using different smooth speed $k_\beta$. Table 10 provides safe state ratio results.

| $k_\beta$ | Ant Hole | Ant North | Ant Range |
|---|---|---|---|
| 3 | 1041.60 ± 226.01 | 1349.00 ± 162.01 | 312.40 ± 59.77 |
| 5 | 1149.20 ± 127.05 | 1333.20 ± 129.10 | 362.20 ± 94.55 |
| 10 | 1068.40 ± 321.47 | 1251.80 ± 99.38 | 399.40 ± 65.12 |
| $\infty$ | 991.60 ± 188.15 | 1120.80 ± 420.01 | 320.00 ± 91.07 |

We also report the safe ratio results for the ablation study in Table 9 and 10.

Table 9: Comparison of the safe state ratio (%) of COMPASS and various query selection methods.

| Method | Ant Hole | Ant North | Ant Range |
|---|---|---|---|
| COMPASS | $90.60 _{\pm 2.40}$ | $96.90 _{\pm 2.50}$ | $80.90 _{\pm 7.80}$ |
| Uniform | $85.80 _{\pm 5.50}$ | $97.00 _{\pm 3.10}$ | $54.40 _{\pm 11.50}$ |
| Uncertainty | $85.80 _{\pm 5.40}$ | $91.70 _{\pm 7.80}$ | $49.90 _{\pm 11.50}$ |

Table 10: Safe state ratio results (%) of COMPASS using different smooth speed $k_\beta$.

| $k_\beta$ | Ant Hole | Ant North | Ant Range |
|---|---|---|---|
| 3 | $90.10 _{\pm 6.00}$ | $96.40 _{\pm 6.20}$ | $77.40 _{\pm 6.00}$ |
| 5 | $90.60 _{\pm 2.40}$ | $96.90 _{\pm 2.50}$ | $80.90 _{\pm 7.80}$ |
| 10 | $89.80 _{\pm 8.10}$ | $92.60 _{\pm 6.40}$ | $83.00 _{\pm 4.30}$ |
| $\infty$ | $91.10 _{\pm 6.30}$ | $92.80 _{\pm 7.70}$ | $77.50 _{\pm 5.40}$ |

### E.3 ADDITIONAL EXPERIMENTS AND ABLATION STUDIES

**Ablation of segment length $H$.** We evaluate COMPASS with varying segment lengths $H$. As shown in the Table 11 and Fig. 4, COMPASS consistently achieves superior performance across different segment lengths ($H = 20, 40, 60$), demonstrating its robustness to this parameter.

Table 11: The safe state coverage results of COMPASS with different segment length $H$.

| $H$ | Ant North | Ant Range |
|---|---|---|
| 20 | $1333.20 _{\pm 129.10}$ | $362.20 _{\pm 94.55}$ |
| 40 | $1325.20 _{\pm 75.70}$ | $309.25 _{\pm 77.00}$ |
| 60 | $1327.60 _{\pm 132.60}$ | $347.00 _{\pm 35.24}$ |

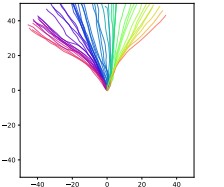 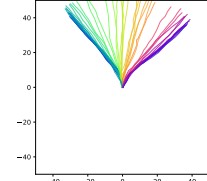 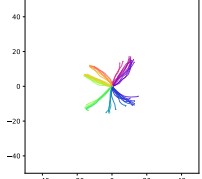 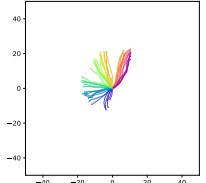

(a) Ant North, $H = 40$.   (b) Ant North, $H = 60$.   (c) Ant Range, $H = 40$.   (d) Ant Range, $H = 60$.

Figure 4: Visualizations of skills learned by COMPASS with varing segment length $H$.

**Comparison to more baselines.** To better show COMPASS's effectiveness, we compare CDP (Hussonnois et al., 2023) with COMPASS. CDP guides the skill discovery by constructing preferred regions, which are identified by a reward model learned from pairwise human preferences. As shown in the Table 12, COMPASS achieves higher safe state ratios, primarily due to its ability to handle sparse data of only 40 labels. The reward model in CDP fails to learn effectively with sparse data, while our training-free guidance method leverages the coherent semantic structure of the latent space, yielding significantly improved performance.

Table 12: Safe state coverage comparison of COMPASS and CDP (Hussonnois et al., 2023).

| Method | Ant North | Ant Range |
|---|---|---|
| **COMPASS** | $\mathbf{1333.20} _{\pm 129.10}$ | $\mathbf{362.20} _{\pm 94.55}$ |
| CDP | $-39.20 _{\pm 14.88}$ | $72.40 _{\pm 1.96}$ |

**Ablation of feedback number $N_{\text{total}}$.** We evaluate COMPASS with a reduced number of samples. As shown in the Table 13 and Fig. 5, COMPASS effectively aligns with human intent even with only 10 or 20 labels, while its performance improves as the number of labels increases.

Table 13: Safe state coverage results of COMPASS using varying number of feedback labels.

| # of labels | Ant North | Ant Hole |
|---|---|---|
| 40 | $1333.20_{\,\pm\,129.10}$ | $1149.20_{\,\pm\,127.05}$ |
| 20 | $1035.60_{\,\pm\,591.51}$ | $829.60_{\,\pm\,243.73}$ |
| 10 | $801.40_{\,\pm\,654.11}$ | $766.20_{\,\pm\,220.39}$ |

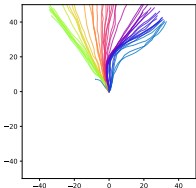 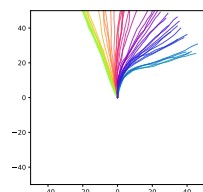 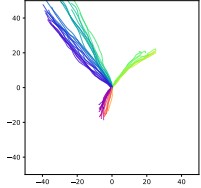 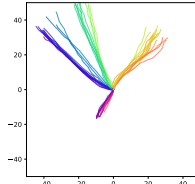

(a) Ant North, $N_{\text{total}} = 20$. (b) Ant North, $N_{\text{total}} = 10$. (c) Ant Hole, $N_{\text{total}} = 20$. (d) Ant Hole, $N_{\text{total}} = 10$.

Figure 5: Visualizations of skills learned by COMPASS with varing feedback number $N_{\text{total}}$.

**Evaluation under noisy feedback.** We evaluate COMPASS under noisy labeling conditions. To simulate human labeling errors, we randomly assign labels (neutral or bad) to states within a band of width $R_{\text{error}}$ around the safety boundaries, reflecting potential human uncertainty in these regions. As shown in the Table 14 and Fig. 6, COMPASS remains robust under noisy labels, indicating the reliability of our training-free guidance mechanism.

Table 14: Safe state coverage results of COMPASS under noisy labels with different levels of noise.

| $R_{\text{error}}$ | Ant North | Ant Range |
|---|---|---|
| 0 | $1333.20_{\,\pm\,129.10}$ | $362.20_{\,\pm\,94.55}$ |
| 0.5 | $1184.60_{\,\pm\,124.84}$ | $369.40_{\,\pm\,51.74}$ |
| 1 | $1084.20_{\,\pm\,135.81}$ | $360.80_{\,\pm\,43.53}$ |

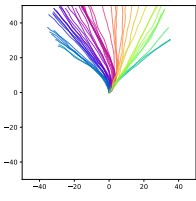 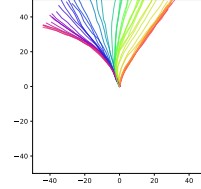 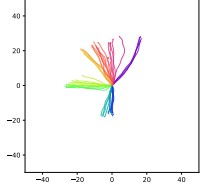 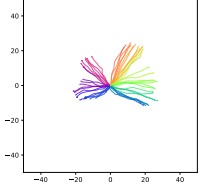

(a) North, $R_{\text{error}} = 0.5$. (b) North, $R_{\text{error}} = 1$. (c) Range, $R_{\text{error}} = 0.5$. (d) Range, $R_{\text{error}} = 1$.

Figure 6: Visualizations of skills learned by COMPASS under noisy feedback with varing $R_{\text{error}}$ on Ant North and Range tasks.

**Human experiments.** We engage human labelers to provide feedback on visualized 2D trajectories in the Ant-Range task, who are instructed by task descriptions in Appendix H.1. We conduct five runs with different seeds, collecting 40 human labels per run. As shown in the Table 15 and Fig. 7, COMPASS consistently achieved high performance, confirming its practical effectiveness with real human input.

Table 15: Safe state coverage results of COMPASS with the oracle teacher in Section 4.1 and with real human labelers.

| Method | Ant Range |
|---|---|
| COMPASS (oracle teacher) | $362.20 _{\pm 94.55}$ |
| COMPASS (human labelers) | $361.40 _{\pm 49.95}$ |

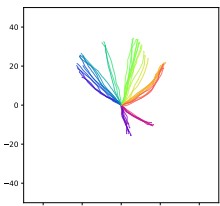

Figure 7: Visualizations of skills learned by COMPASS with real human labelers on the Ant Range task.

**Evaluation on increasing skill latent dimension.** To investigate whether under-explored regions, such as areas behind hole hazards in the Ant Hole task, result from the limited representational capacity of the skill space, we increased the skill latent dimension from 2 to 4 to enhance its ability to encode diverse behaviors. However, as visualized in Fig. 8, the expanded skill space did not lead to exploration of the occluded regions. The underlying reasons for the formation of under-explored regions are further analyzed in Appendix F.

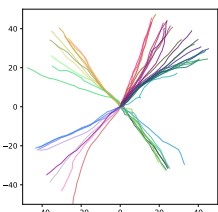

Figure 8: Visualizations of skills learned by COMPASS using 4-dim skills.

**Validation on the semantically coherent latent space.** To demonstrate the importance of the semantical coherence property in COMPASS, achieved by the temporal distance constraint, we compare COMPASS with its variant, which uses the Euclidean distance between raw states as the distance constraint in the DSD framework, i.e. $d(x,y) = \|x - y\|_2$ in Eq. 1. As shown in the Table 16, using Euclidean distance performs worse than COMPASS, which uses the semantically coherent latent space. This highlights the importance of semantically coherent latent representations.

Table 16: Comparison of the safe state coverage results of COMPASS using temporal distance and Euclidean distance as the distance constraint.

| | Ant North | HalfCheetah Not-Flip |
|---|---|---|
| COMPASS | $1333.20 _{\pm 129.10}$ | $215.60 _{\pm 3.05}$ |
| COMPASS (with Euclidean distance) | $657.60 _{\pm 432.54}$ | $104.20 _{\pm 21.51}$ |

## F    DISCUSSION AND LIMITATION

**Dead zone phenomenon in scenarios with obstacles.**    Despite the COMPASS's strong performance, we identified under-explored regions in complex scenarios like "Hole". As shown in Fig. 2, both COMPASS and DoDont* fail to explore areas behind the holes in the Ant Hole task. Even with accurate guidance, as demonstrated by the "Oracle" results, the base skills fail to bypass obstacles and reach areas hidden behind them.

We believe this is primarily due to the inherent exploration mechanisms of the underlying METRA framework. In scenarios with obstacles, METRA's optimal behaviors do not encourage skills to explore regions behind obstacles, resulting in "dead zones".

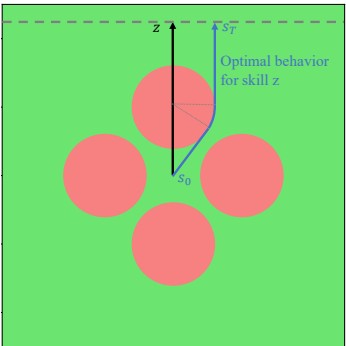 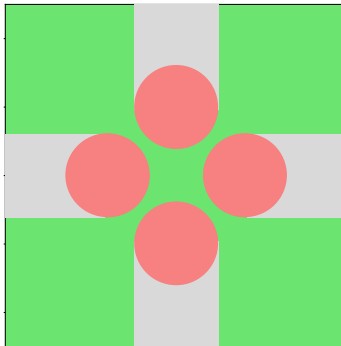

Figure 9: Visualization of the reason for the dead zone phenomenon in scenarios with obstacles (Hole as an example). Human-undesirable regions are highlighted in red, and other regions are highlighted in green. In the right subfigure, gray regions represent "dead zones".

We use Fig. 9 to illustrate this phenomenon. Consider a mass point in a "Hole" scenario, with a 2-dimensional skill space, skills are expected to move in diverse directions. If we ignore the human-undesirable regions, the METRA's latent space $\phi(s)$ aligns with the 2D state space, i.e., $\phi(s) = s$. Since COMPASS's objective (Eq. 10) differs from METRA only by reweighting, if the samples to train the latent space are sufficient and the non-human-undesirable regions are well explored, the latent space COMPASS learned will be equivalent to METRA learned in the non-human-undesirable regions.

Consequently, for the specific skill latent $z$ shown in Fig. 9, the optimal behavior follows the blue trajectory, which achieves the largest reward $(\phi(s') - \phi(s))^T z$ within the fixed timesteps. The trajectory of such optimal behavior will go parallel with the skill latent $z$ after passing the human-undesirable regions, which makes the region just behind the hole a dead zone.

The main reason for the dead zone is that METRA uses the DSD framework only to encourage the coverage of skills in the state space by aligning the trajectory with the uniformly distributed skill latent $z$, without using any pure exploration schemes such as prediction errors (Pathak et al., 2017; Burda et al., 2019), state entropy (Hazan et al., 2019; Liu & Abbeel, 2021b), or pseudo-counts (Bellemare et al., 2016; Ostrovski et al., 2017). This makes METRA easily ignore the under-explored areas, even if they are close to the learned skills and are not hard to reach. Combining the DSD framework with exploration schemes is a promising aspect for addressing the dead zone phenomenon.

**Bound tightness of Proposition 1.**    A limitation of the current work is that the proposed bound is not uniformly tight across the state space, especially near the transition point of $g(s)$, where the Bayes error rate $P^*(s)$ could be high. However, in the current work, the primary purpose of the proposed bound is to justify the construction of our guidance signal. In regions of the state space far from transition boundaries, where the Bayes error rate $P^*(s)$ is small, the bound is non-trivial and ensures the performance. Also, while the bound is looser near transition boundaries, experimental results indicate this does not significantly affect our method's performance. Strengthening the theoretical foundations of COMPASS remains an important direction for future research.

Apart from theoretical analysis, a potential way to mitigate this issue is to use the "neutral" labels as a buffer zone between "good" and "bad" labels. The boundaries between "good" and "neural" labels, as well as between "neural" and "bad" labels, are sufficiently distant from the original "good-bad" boundary (assuming no neural labels), which mitigates the impact of the loose bounds on distinguishing good and bad samples.

**Connection with Diverse Density (DD) and multi-instance learning (MIL).** The construction of the guidance signal in COMPASS shares similarities with multi-instance learning (MIL) (Carbonneau et al., 2018). Specifically, in COMPASS, we construct a state-level guidance signal for skill discovery with segment-level human feedback. This approach resembles MIL, which utilizes bag-level annotations to address uncertainty in instance-level labels. Moreover, the computation of COMPASS's guidance signal $w(s)$ is based on the distance between states in the latent space and the annotated state set, which is conceptually similar to the distance-weighted probability modeling in the noisy-or method of Diverse Density (DD) (Maron & Lozano-Pérez, 1997), a well-known MIL method.

Due to the similarities, a straightforward idea is that integrating DD with our method could potentially improve performance for long segments, because DD enables finer-grained utilization of labeled samples, eliminating the need for all samples within the same segment to share identical labels, as currently required in COMPASS. However, DD has limitations in directly addressing the three-class problem (good/neutral/bad). Also, its reliance on concept points to model probabilities assumes clustering of positive and negative samples (Carbonneau et al., 2018), and requires a predefined number of concept points, which may not suit tasks like Ant Hole in our study. Moreover, DD and related multi-label learning (MIL) methods involve optimization processes, which may contradict COMPASS's key advantage of avoiding auxiliary model training.

As the aforementioned challenges could not be easily addressed, we did not adopt DD in our current work. Nevertheless, given the similarities between DD and our method, we believe that DD and MIL offer alternative perspectives for understanding our method. This provides a promising aspect for further improvement and analysis of our method.

**Why successive states along a trajectory probably share similar desirability.** While unsupervised methods based on pure exploration methods (e.g., state entropy) or random trajectories might exhibit the behaviour that repeatedly crossing the negative region, such behavior does not occur in our approach due to the DSD backbone, where the learned skills inherently follow relatively straight paths, as this maximizes the objective function $(\phi(s_T) - \phi(s_0))^T z$.

Specifically, skills that repeatedly cross a safety boundary tend to have smaller distances between the initial state and the final state within a fixed horizon $T$. In contrast, moving along a straighter trajectory results in a significantly larger distance along the skill direction, resulting in larger cumulative intrinsic rewards $(\phi(s_T) - \phi(s_0))^T z$.

As a result, such behaviors that repeatedly cross a safety boundary are neither converged nor optimal, while optimal policies tend to follow straighter trajectories. This is further verified by our experimental results (as shown in Fig. 2), where visualized trajectories show no repeated crossings.

**Properties of guidance for high-quality skill learning.** As demonstrated in the ablation studies in Section 4.4, both the distribution of the high-quality guidance signal and the scheduling of guidance signal inclusion are essential for effective skill learning. We elaborate on these two aspects below.

The distribution of the high-quality guidance signal is mainly impacted by the distribution of labeled states, as our guidance signal construction method relies on the nearest neighbor approach. Therefore, sufficient coverage of the high-quality guidance signal across the state space is essential for effective skill learning. This requires that the labeled states are sufficiently scattered. Apart from encouraging sufficient coverage directly (through entropy-based sampling during query selection, as is used in COMPASS), we also explored alternative strategies for distributing high-quality guidance signals. Specifically, we considered reducing the uncertainty of the guidance signal using uncertainty-based sampling during query selection (referred to as Uncertainty in Table 7), and adopting a random sampling approach (referred to as Uniform in Table 7). As shown in Table 7, both the Uncertainty method and the Uniform method perform significantly worse than COMPASS,

showing the significance of achieving sufficient coverage of high-quality guidance signals across the state space for effective skill learning.

For the scheduling of guidance signal inclusion, gradually introducing the guidance signal into the USD is essential for effective skill learning. We have tried to introduce the guidance signal abruptly into the USD. However, this significantly degraded the performance. As shown in Table 8, pure GSD performs the worst. The primary reason is that at the beginning of training, the algorithm employs USD to gather data, distributing skills across the state space. However, when USD is abruptly switched to GSD, some skills become trapped within the center of human-undesirable areas. In these regions, skills receive zero rewards and are almost impossible to escape with the limited exploration mechanism in DSD.

## G  EXTENDED RELATED WORK

**Unsupervised reinforcement learning.**   Unsupervised reinforcement learning (Xie et al., 2022) learns a policy or set of policies with unlabeled data (transitions without task-specific rewards) to explore the state space. The target is to acquire knowledge of the environment, thereby facilitating downstream tasks. To encourage exploration, various intrinsic rewards are proposed, such as prediction errors (Pathak et al., 2017; Burda et al., 2019), state entropy (Hazan et al., 2019; Liu & Abbeel, 2021b), pseudo-counts (Bellemare et al., 2016; Ostrovski et al., 2017), and empowerment measures (Eysenbach et al., 2019; Sharma et al., 2020).

**Unsupervised skill discovery.**   Skill discovery methods construct intrinsic reward with empowerment measures and learn a set of distinguishable policies to cover the state space jointly. A typical choice for the empowerment measure is the mutual information $I(s, z)$ between the state $s$ and the skill latent $z$. Recent studies explore different mutual information formulations. The reverse form $I(s, z) = H(z) - H(z|s)$ (Eysenbach et al., 2019; Park et al., 2022b) trains an additional skill discriminator $q(z|s)$ to encourage skills to visit different states. The forward form $I(s, z) = H(s) - H(s|z)$ (Sharma et al., 2020; Liu & Abbeel, 2021a; Laskin et al., 2022) trains an additional state density model $q(s|z)$ for each skill, enabling integration with model-based RL algorithms. Though maximizing the mutual information could induce diverse behaviors, this does not encourage exploration, which may lead to static behaviors (Park et al., 2022b; 2023b).

To address this issue, recent studies have explored various strategies, such as employing exploration methods to collect diverse trajectories (Campos et al., 2020), incorporating an entropy maximization term into the intrinsic reward (Liu & Abbeel, 2021a), and eliminating the anti-exploration term from mutual information in skill learning (Zheng et al., 2025). An outstanding category is the Distance-maximizing Skill Discovery approach (DSD) (Park et al., 2023a), which links the distance in latent space with that in space to encourage coverage in state space. The objective function of DSD is formally derived in METRA (Park et al., 2023b) by replacing the traditional mutual information objective in SD with the Wasserstein dependency measure (WDM). The distance could be any arbitrary function $d(\cdot, \cdot) : \mathcal{S} \times \mathcal{S} \to \mathbb{R}_0^+$ to encourage exploring state sub-space with different properties. For example, Euclidean distance (Park et al., 2022b) encourages geometrically longer travel, the negative log-likelihood of an estimated transition probability (Park et al., 2023a) encourages visiting rarely visited states, and temporal distance (Park et al., 2023b) encourages temporally far travel. However, these DSD methods do not consider human desirability when exploring, which makes the exploration inefficient when the state space is vast and complex.

**Guided skill discovery.**   Recent studies mitigate unnecessary exploration in unsupervised skill discovery by incorporating prior knowledge into skill learning. The prior knowledge could come from expert trajectories (Klemsdal et al., 2021; Kim et al., 2024) and analytical formulas of constraints (Kim et al., 2023). Specifically, Klemsdal et al. (2021) and DoDont (Kim et al., 2024) train a classifier to distinguish expert trajectories from other trajectories. Klemsdal et al. (2021) further uses the encoder of the classifier as a state projection to encourage exploring the expert-concerned state subspace. While DoDont use the probability output by the classifier as the distance function in DSD. Kim et al. (2023) considers Lagrangian Q learning in skill learning to ensure the safety of learned skills. Recent studies explore utilizing pairwise human preferences (Hussonnois et al., 2023; 2025) to learn a human-aligned reward model. These models then guide the skill discovery process by identifying preferred regions (Hussonnois et al., 2023) or encouraging alignment between skills and human values (Hussonnois et al., 2025). Despite these advancements, deriving expert trajectories or constraint formulas is often challenging and impractical in complex tasks, and classifiers or reward models trained on limited data can be unstable. This paper aims to address these limitations.

**Human feedback in policy learning.**   Prior works have demonstrated the effectiveness of integrating human feedback into policy learning to overcome the challenges of manual reward design. Early works such as Daniel et al. (2014) used active queries with numerical ratings, while Akrour et al. (2014) and Sugiyama et al. (2012) leveraged pairwise preferences to iteratively refine policies or reward functions. Wang et al. (2016) explored interactive learning mechanisms in language games through implicit selection feedback. Building on these foundations, preference-based reinforcement learning (PbRL) (Christiano et al., 2017) has emerged as a key framework for aligning agents with

human intent through structured comparisons. Inspired by these efforts, COMPASS incorporates human guidance to direct exploration toward desirable behaviors.

However, a core limitation of PbRL is the high cost of human supervision. To address this issue, recent PbRL studies have focused on improving feedback efficiency via enhanced query selection (Lee et al., 2021; Shin et al., 2023), unsupervised pretraining (Lee et al., 2021; Cheng et al., 2024), and data augmentation (Park et al., 2022a; Choi et al., 2024). Despite these efforts, recent studies show that pairwise comparisons, a common approach in PbRL, suffer from segment indistinguishability (Mu et al., 2025), which significantly undermines their effectiveness. Since feedback is scarce in our work, and the pretraining phase involves numerous potential tasks, which makes the indistinguishability issue more severe, COMPASS adopts discrete ratings for single segments, similar to (Akrour et al., 2014) and (Sugiyama et al., 2012), to avoid the indistinguishability problem inherent in pairwise comparisons. Additionally, unlike standard approaches that require training parameterized reward models, COMPASS proposes a training-free method that efficiently utilizes sparse feedback by leveraging the semantic coherence of the unsupervised skill latent space.

# H EXPERIMENTAL DETAILS

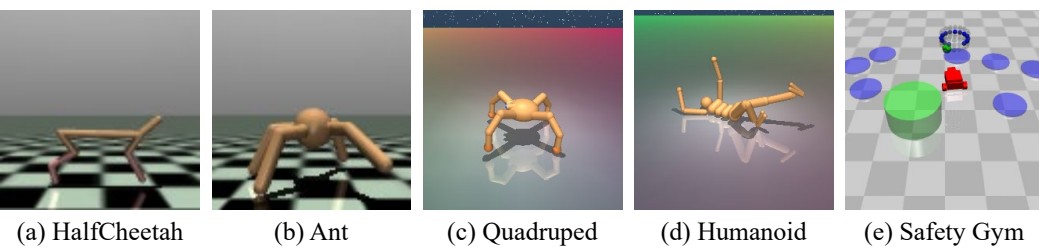

(a) HalfCheetah        (b) Ant        (c) Quadruped        (d) Humanoid        (e) Safety Gym

Figure 10: Benchmark environments.

## H.1 SETUP

**Environments.** We evaluate COMPASS on five complex robotic locomotion environments: state-based Ant and HalfCheetah from OpenAI Gym (Todorov et al., 2012; Brockman et al., 2016), pixel-based Quadruped and Humanoid from DMControl (Tassa et al., 2018), and state-based Safety Gym (Ray et al., 2019), as illustrated in Fig. 10. In pixel-based DMControl environments, we follow prior works (Park et al., 2022b; 2023b; Kim et al., 2024) by using colored floors, allowing the agent to infer its location from pixel observations. The observation is $64 \times 64$ RGB images of the scene for these pixel-based environments. For Safety Gym (Ray et al., 2019), we use a customized `Safexp-CarGoal1-v0` environment, where a car must navigate to a goal while avoiding hazards. To ensure consistency within a single experiment, the locations of hazards are randomly generated at the start of each experiment but remain fixed throughout its duration.

**Guidance task designs.** To assess COMPASS's alignment with human intent, we design tasks with varying guidance types:

- **Direction guidance**, where the agent moves towards a specific direction (*North* and *Right*).
- **Range guidance**, where the agent explores within a range (*Range*).
- **Hazard avoidance**, where the agent avoids hazardous areas (*Hole* and *Hazard*).
- **Unsafe behaviour avoidance**, where the agent avoids unsafe actions (*Not-Flip*).
- **Composite tasks**, which combine multiple guidance types, requiring hazard avoidance while encouraging directional movement (*Range-North* and *Hole-North*).

Tasks are illustrated in Figure 2. The oracle guidance signals for states are specified as follows:

- **North** (for Ant and Quadruped environments): A state is considered bad if the location $(x, y)$ does not satisfy $y \geq |x|$.
- **Right** (for HalfCheetah environment): A state is considered bad if the location $x$ does not satisfy $x \geq 0$.
- **Range** (for Ant environment): There is a safe area defined as a circle with its center at $(0, 5)$ and radius $r = 20$. A state is considered bad if the agent is outside this safe area.
- **Hole** (for Ant and Humanoid environment): There are four holes in the scene. A state is considered bad if the agent is located in any of these holes: For Ant environment, the holes are circles with centers at $(0, 20)$, $(0, -20)$, $(20, 0)$, $(-20, 0)$, all with a common radius $r = 12$. For Humanoid environment, the holes are circles with centers at $(0, 8)$, $(0, -8)$, $(8, 0)$, $(-8, 0)$, all with a common radius $r = 4$.
- **Hazard** (for Safety-Gym environment): A state is considered bad if the agent is in either hazard area. The locations of hazards are randomly generated at the start of each experiment but remain fixed throughout its duration.
- **Not-Flip** (for HalfCheetah environment): A state is considered bad if the agent flips. Specifically, the agent is said to have flipped when the absolute value of its pitch angle exceeds 90 degrees.

- **Range-North** (for Ant environment): There is a safe area defined as a circle with center at $(0, 5)$ and radius $r = 20$. A state is considered bad if the agent is outside this safe area. Additionally, if the agent is in the safe area and the location $(x, y)$ satisfies $y \geq |x|$, then the state is considered good.

- **Hole-North** (for Ant environment): There are four holes in the scene, defined as circles with centers at $(15, 15)$, $(15, -15)$, $(-15, -15)$, $(-15, 15)$, and a common radius $r = 12$. A state is considered bad if the agent is in either hole. Additionally, if the agent is not in any of the holes and the location $(x, y)$ satisfies $y \geq |x|$, then the state is considered good.

**Metrics.** We employ three main metrics for evaluation:

- **Safe state coverage**, which measures the agent's ability to explore the state space while avoiding hazardous regions. Following (Kim et al., 2024), this metric assigns a value $+1, -1$ to safe and unsafe areas, and computes state coverage by counting the unique $1 \times 1$ x-y bins (or 1-unit x-axis bins for HalfCheetah tasks) visited by the agent. For Safety-Gym tasks, we set the x-y bin size to $0.01 \times 0.01$ due to the small scale of the coordinates.

- **Safe state ratio**, which quantifies the proportion of visited safe bins among all visited bins. It is defined as the ratio of the number of unique safe bins (labeled as good or neutral) to the number of unique unsafe bins (labeled as bad).

- **Downstream task performance**, which evaluates the utility of learned skills in downstream tasks. We consider both a zero-shot setting and a task-specific hierarchical control setting. To assess zero-shot performance, we roll out the downstream task environment with randomly sampled skills and report both the average and the best performance across all sampled skills. To assess hierarchical control performance, we use the learned skills as a low-level controller and train an additional high-level controller to optimize performance on the downstream task. The downstream task performance is then reported as the hierarchical control performance of the pretrained skills. Further details on the downstream tasks can be found in Appendix H.2, while additional information about the high-level controller is provided in Appendix H.3.

## H.2 DOWNSTREAM TASK DETAILS

**Zero-shot performance.** We evaluate the zero-shot performance of the learned skills (in Ant tasks) on a customized Ant motion task. The single-step reward comprises a survival reward (1.0 per step), a movement reward (the maximum of the forward speed and the lateral speed), and a safety penalty ($-20.0$ if the agent enters the unsafe area defined by the guidance task used during skill learning).

**Hierarchical control performance.** We evaluate the hierarchical control performance of the learned skills (in HalfCheetah Not-flip tasks) on a HalfCheetah Goal task. The agent will receive a reward of 1.0 if it is sufficiently close to the goal (i.e., within a distance of less than 3), and a safety penalty of $-20.0$ if the agent flips. The goals are randomly sampled from the range $[-100, 100]$.

## H.3 IMPLEMENTAL DETAILS

We implement METRA on top of the publicly available CSF codebase[1] (Zheng et al., 2025), as it provides more detailed scripts and supports the evaluation of downstream task performance. An anonymous code repository is provided:



https://anonymous.4open.science/r/SKILLCOMPASS



For the baselines, we adopt the implementation of METRA (Park et al., 2023b), DIAYN (Eysenbach et al., 2019), and LSD (Park et al., 2022b) from the CSF codebase, and implement Online DoDont (DoDont*) on top of the CSF codebase. For CDP (Hussonnois et al., 2023), we use their official codebase.[2]

---

[1]https://github.com/Princeton-RL/contrastive-successor-features

[2]https://github.com/HussonnoisMaxence/CDP

COMPASS shares the same hyperparameters as the baselines, which are consistent with METRA. We list these hyperparameters in Table 17.

Table 17: Common hyperparameters for unsupervised skill discovery methods.

| Hyperparameter | Value |
| --- | --- |
| Encoder for pixel tasks | CNN (LeCun et al., 1989) |
| # hidden layers | 2 |
| # hidden units per layer | 1024 |
| Learning rate | 0.0001 |
| Optimizer | Adam (Kingma & Ba, 2014) |
| Minibatch size | 256 |
| Target network smoothing coefficient | 0.995 |
| Entropy coefficient | auto-adjust (Haarnoja et al., 2018) |
| Total horizon length | 200 |
| # episodes per epoch | 8 |
| # gradient steps per epoch | 200 (Quadruped, Humanoid), 50 (Ant, HalfCheetah) |
| Discount factor $\gamma$ | 0.99 |
| METRA $\epsilon$ | $10^{-3}$ |
| METRA initial $\lambda$ | 30 |

The additional hyperparameters of COMPASS are listed in Table 18.

Table 18: Additional hyperparameters for COMPASS.

| Hyperparameter | Value |
| --- | --- |
| Segment length | 20 |
| Feedback frequency | 1000 |
| Warm-up epochs before the first feedback | 2000 |
| The total feedback amount | 40 (Ant, HalfCheetah, Safety-Gym), 100 (Quadruped, Humanoid) |
| The feedback amount per session | 5 |

For hierarchical control tasks, we use a PPO (Schulman et al., 2017) agent as the high-level controller. The trained skills serve as the low-level controller, and their parameters are fixed during the training of the hierarchical control agent. The hyperparameters are shown in Table 19.

Table 19: Hyperparameters for high-level controllers.

| Hyperparameter | Value |
| --- | --- |
| Learning rate | 0.0001 |
| Option timesteps length | 25 |
| Total horizon length | 200 |
| Replay buffer batch size | 256 |
| # hidden layers | 2 |
| # hidden units per layer | 1024 |
| Temperature $\alpha$ | 1 |

