# OpenReview forum: "COMPASS: Training-Free Guidance for Skill Discovery with Human Feedback"
_ICLR.cc/2026/Conference — Submitted to ICLR 2026_

### Official Review · Reviewer_tmQR · 2025-10-25

**Soundness:** 2
**Presentation:** 2
**Contribution:** 3
**Rating:** 2
**Confidence:** 4

**Summary:**

Extends guided skill discovery with positive and negative labels to a small number of states to group skills in latent space according to similar human desirability without training. These skills are theoretically reliable in the sense that the probability of the skills failing is dependent on the smoothness of the feedback in latent space. Then, the skills avoid hazardous behaviors and achieve superior downstream performance empirically in simple mujoco domains with negative regions.

**Strengths:**

This work introduces a simple and elegant method for including trinary human feedback into the online learning process of unsupervised skill discovery. This setting is underexplored and the simplicity is valuable.

This method provides an evaluation of the method and demonstrates that under appropriate settings it can perform well.

**Weaknesses:**

This work tends to overstate and obfuscate what appears to be a straightforward concept and application, which makes it harder rather than easier to understand.

The evaluations for a setting predicated on human feedback signals do not include a human element such as noise, ease of providing feedback, or human evaluations.

The theoretical bounds appear to be quite loose in settings that are plausible, so the efficacy of the method is predicated on assumptions that are not necessarily strong, and there are no visualizations to suggest that a simpler metric, such as using the Euclidean distance between raw states would not work just as well in the settings evaluated.

**Questions:**

Why isn't the insight of how the semantics are included (human trinary feedback) part of the abstract? This seems like an addition that would make it much less vague, since that appears to be the main algorithmic contribution and requirement for the method.

It seems like a formal description of guided skill discovery is missing from the preliminaries and perhaps not well defined, and since this is a core part of the algorithm, this is a significant oversight.

The use of the term "training free" seems to be inaccurate. While the feedback does not need to be separately trained, it does have to be incorporated during training. Thus, the term training-free is misleading, since without incorporation into the training loop the guidance signal is useless (unlike a method which modifies an existing set of unsupervised skills training free to match the desired constraints).

The derivation of the "equivalent but more practical objective" is given no space in the main paper. At least some intuition for how this transformation is justified in the main paper would be useful.

The notion of semantic coherence seems at odds with the notion that "states occurring successively along a trajectory typically share similar desirability" (L208). Besides the fact that "typically" is imprecise, it seems like a trajectory drawn from an unsupervised algorithm could easily move in and out of desirability, such as if it were desirable to move toward a goal, and the agent moved alternately towards and away from the goal (or in and out of a negative region). If the trajectories are sampled from an expert, then this would need to be stated since that is a significant departure from USD.

A small note, but the use of "camera rotation in visual environments" is not a good example where semantically similar states may be far apart in state space, since camera rotation is not really a transformation of state as much as it is a transformation of observation in a partially observed environment.

While the Bayes error rate is useful, it is not clear from the main paper how tight this bound is, since it seems like this value could actually be quite large in the case when the state is near a transition point of g(s).

The description from the introduction seems overstated, considering that the method is simply reweighting the DSD objective with a soft nearest neighbor in latent space.

Because the algorithm is not included in the main paper, Algorithm 1 seems like it would be necessary to be included in the main paper; otherwise, the actual procedure of COMPASS is opaque (the core algorithm should not be buried in the appendix)

Since this guidance is used during training, it seems like there could be a case where the choice of guidance makes it hard for the agent to learn. In fact, this appears to be at least somewhat the case in the "guidance signal smoothing" section. A greater discussion of this and the actual techniques that were tried and failed should be included in the Appendix, since this would be a core tool for a downstream use case.

The evaluations seem insufficient, since one would expect a performance drop in some domains by adding the human constraint, but because of the choice of evaluations as simply 2D positive and negative regions with clear gaps for exploration, this does not end up being applied in a meaningful way. Inclusion of both noisy "human signals" and human feedback which transforms the task to be hard exploration, would make this a much more effective evaluation.

Another inadequacy of this work is the lack of human signals. Because this work is marketed as being guided by human segmentation, there is an expectation that at least some experiments where a human provides the guidance signal would be necessary to illustrate that the method has grounding.

The choice of baselines seems inadequate, since most of the methods are not meant to include human guidance at all (except for DoDont, which appears to be applied to a different setting). Since there appears to be a wider range of GSD algorithms, it would be useful to include more of these.

It seems like a good deal of coverage is left unreached in the experiments, so experiments with a much greater number of skills would help illuminate if the feedback produces dead regions, where, because of the feedback, the agent does not learn to reach certain positive regions, and if other methods can reach those regions.

---

> ### Author Response · Authors · 2025-11-21
> **Response to Reviewer tmQR: Part 1**
>
> Dear Reviewer,
>
> We sincerely thank you for the valuable and detailed feedback, as well as the opportunity to clarify our contributions.
> We hope the following statement clears your concern.
>
>
> **W1, Q1, Q8: Contribution concerns.**
>
> **A for W1, Q1, Q8:**
> The reviewer suggests that COMPASS primarily focuses on "integrating good/neutral/bad human feedback into the online learning process of unsupervised skill discovery" and "reweighting the DSD objective with a soft nearest neighbor in latent space".
> We respectfully clarify that these are not the core contributions of our work.
>
> The primary contributions of COMPASS are as follows:
>
> 1. We propose a **training-free method for constructing guidance signals**. Leveraging the semantically coherent latent space, the constructed guidance signal is reliable with minimal labels, without the need for auxiliary model training.
> In contrast, prior works [1, 2] require learning a guidance signal model before skill discovery, which is time-consuming and prone to overfitting on limited data, thus being unreliable.
> 2. We provide a **detailed analysis to justify our GSD framework**, showing the necessity of a semantically coherent latent space for training-free guidance signal construction.
> This analysis supports our use of METRA's temporal distance, ensuring the soundness of our approach. Additionally, it allows for the selection of other semantically coherent DSD methods beyond METRA, ensuring future extensibility.
> 3. We propose an **active query selection method** to gather human feedback, enabling adaptive selection of high-quality data during skill training.
> In contrast, prior work [1] relies on pre-collected human-labeled trajectories, making its performance heavily dependent on dataset quality, introducing challenges in dataset construction.
>
> We thank you for your detailed comment and hope this response clarifies the unique contributions of our work.
>
> **W2, Q11, Q12: More evaluations under practical human feedback signals.**
>
> **A for W2, Q11, Q12:**
> As suggested, we conduct two additional experiments to evaluate COMPASS (1) under noisy labeling conditions and (2) with real human labelers.
>
> 1. **Noisy labeling experiments**:
> We simulate human labeling uncertainty by randomly assigning labels to states within a band of width $R_\text{error}$ around the safety boundaries, reflecting potential human uncertainty in these regions.
> As shown in the table below and visualized in Fig. 6 (Appendix E), COMPASS maintains superior safe state coverage, demonstrating the robustness of our training-free guidance signal even with imperfect labels.
>
>
> | $R_\text{error}$ | Ant North | Ant Range |
> | :-: | :-: | :-: |
> | 0 | 1333.20 ± 129.10 | 362.20 ± 94.55 |
> | 0.5 | 1184.60 ± 124.84 | 369.40 ± 51.74 |
> | 1 | 1084.20 ± 135.81 | 360.80 ± 43.53 |
>
>
> 2. **Human experiments**:
> We engage human labelers to provide feedback on visualized 2D trajectories in the Ant-Range task, following task descriptions.
> We conduct five runs with different seeds, collecting 40 human labels per run.
> As shown in the table below and visualized in Fig. 7 (Appendix E), COMPASS consistently achieved high performance, confirming its practical effectiveness with real human input.
>
> | | Ant Range |
> | :-: | :-: |
> | COMPASS (oracle teacher) | 362.20 ± 94.55 |
> | COMPASS (human labelers) | 361.40 ± 49.95 |

---

> ### Author Response · Authors · 2025-11-21
> **Response to Reviewer tmQR: Part 2**
>
> **W3, Q7: Concerns on theoretical bounds and the necessity of the semantically coherent latent space.**
>
> **A for W3, Q7:**
>
>
> We thank the reviewer for raising this crucial point about bound tightness and the significance of the semantically coherent latent space.
>
> **(1) Bound tightness.**
> We acknowledge that the bound is not uniformly tight across the state space, particularly near the transition points of $g(s)$. However, its primary purpose is to validate the construction of our guidance signal.
> In regions of the state space far from transition points, where $P^*(s)$ is small, the bound is non-trivial and ensures the performance.
> Also, while the bound is looser near transition points, experimental results indicate this does not significantly affect our method's performance.
>
> We recognize this as a limitation and have included it in Appendix F, with plans to address it in future work.
> A potential way to mitigate this issue is to use the neural label as a buffer zone between "good" and "bad" labels.
> The boundaries between "good" and "neural" labels, as well as between "neural" and "bad" labels, are sufficiently distant from the original "good-bad" boundary (assuming no neural labels), thereby mitigating the impact of the loose bounds on distinguishing good and bad samples.
>
> **(2) Validation on the semantically coherent latent space.**
> As suggested, we conduct additional experiments using the Euclidean distance between raw states as the distance constraint in the DSD framework, that is,
> $$
> ||\phi(x) - \phi(y)||_2 \le ||x-y||_2, \, \forall x,y \in \mathcal{S}
> $$
> As shown in the table below, using Euclidean distance performs worse than COMPASS, which uses the semantically coherent latent space.
> This highlights the importance of semantically coherent latent representations.
>
>
> | | Ant North | HalfCheetah Not-Flip |
> | :-: | :-: | :-: |
> | **COMPASS** | **1333.20 ± 129.10** | **215.60 ± 3.05** |
> | Euclidean distance | 657.60 ± 432.54 | 104.20 ± 21.51 |
>
>
> **Q2: Formal description of guided skill discovery.**
>
> **A for Q2:**
> Thank you for this valuable feedback.
> We agree that a formal definition of Guided Skill Discovery is necessary.
> We have added it in the Preliminaries section (Section 2) to provide a rigorous foundation for our contribution.
>
> > In complex scenarios, USD can be inefficient, as many of the learned skills may be irrelevant or even harmful to downstream tasks.
> > GSD addresses this issue by incorporating human intent to guide skill learning toward desirable behaviors while avoiding undesirable ones.
> > A typical objective is:
> > $$
> > \sup_{\pi,\phi} J_{\text{USD}}(\pi,\phi) + \lambda_{\text{guide}} \cdot {J_{\text{guide}}(\pi,\phi)}\quad \text{s.t.}\quad C_\text{USD}(\phi)\le 0,\quad \lambda_{\text{guide}}^c\cdot C_{\text{guide}}(\phi)\le 0,
> > $$
> > where $J_{\text{USD}}(\pi,\phi)$ represents the USD objective, such as mutual information $I(s, z)$ or the DSD objective.
> > $C_\text{USD}(\phi)$ denotes constraints in USD objectives, such as those in DSD.
> > $J_{\text{guide}}(\pi,\phi)$ reflects the guidance objective derived from human intent, which may include expert trajectories or pairwise human preferences.
> > $C_\text{guide}(\phi)$ is the guidance in the form of constraints, like analytical constraint formulas for safety.
> > The coefficients $\lambda_{\text{guide}},\lambda_{\text{guide}}^c \geq 0$ adjust the strength of guidance.
>
>
> **Q3: The term "training-free" seems to be inaccurate.**
>
> **A for Q3:**
> Thank you for the valuable comment.
> We would like to clarify that in our work, "training-free" specifically refers to the construction of the **guidance signal** $w(s)$, which is directly derived from sparse human labels within a semantically coherent latent space, without requiring any auxiliary model training.
> However, as stated in our objective (Eq. 10-13), the skill policy itself is trained via reinforcement learning.
>
> The term "training-free guidance" emphasizes our key contribution: COMPASS **directly and effectively** incorporates human intent, eliminating the need for expert data collection [1] or additional model training in [1, 2], while preserving the stability and scalability of the underlying skill discovery method.
>
> In response to your comments, we have added further clarification on the concept of training-free guidance in the abstract.
>
>
> **Q4: Explanation for the optimization problem transformation.**
>
> **A for Q4:**
> Thank you for your thoughtful question.
> This derivation is based on a variable substitution, where we replace the latent function with a scaled version, $\phi^{\prime}(s) = \frac{\phi(s)}{w(s)}$.
> By substituting $\phi^{\prime}(s)$ into the original objective (Eq. 9) and utilizing the continuity of $w(s)$, the equivalent form (Eq. 10) is obtained.
> As suggested, we have added a detailed explanation of this substitution process to the main text.

---

> ### Author Response · Authors · 2025-11-21
> **Response to Reviewer tmQR: Part 3**
>
> **Q5: The notion of semantic coherence seems at odds with the notion that "states occurring successively along a trajectory typically share similar desirability".**
>
> **A for Q5:**
> Thank you for the insightful comment.
>
> - **Word choice**: As suggested, we replaced "typically" with "probably" in L208 to enhance precision and better align with the definition in L210.
> - **Impossibility of trajectories repeatedly crossing the negative region**:
> While normal unsupervised methods (pure exploration) or random trajectories might exhibit such behavior, this does not occur in our approach due to the DSD backbone, where the learned skills inherently follow relatively straight paths.
> Specifically, skills that repeatedly cross a safety boundary tend to have smaller distances between the initial state and the final state within a fixed horizon $T$.
> In contrast, moving along a straighter trajectory results in a significantly larger distance along the skill direction, resulting in larger cumulative intrinsic rewards $(\phi(s_T)-\phi(s_0))^Tz$.
> Thus, such behaviors are neither converged nor optimal, while optimal policies tend to follow straighter trajectories.
> This is further verified by our experimental results (Fig. 1 in the main paper), where visualized trajectories show no repeated crossings.
> - **Trajectory source**: The trajectories provided to the labeller are those sampled during the policy training, rather than expert trajectories.
>
>
> **Q6: The use of "camera rotation in visual environments" is not a good example where semantically similar states may be far apart in state space.**
>
> **A for Q6:**
> Thank you for the insightful feedback.
> To better clarify this point, we have revised the text with a more relevant example:
> In robotic locomotion, a robot at a given $(x, y)$ position can be in either a stable (human-desirable) or fallen (undesirable) state.
> Although these states may be very close in the state space, e.g., differing only in joint angles or orientation, they exhibit completely different human desirability.
> This illustrates that **the original state space lacks semantic coherence**, as nearby states can correspond to significantly different levels of human desirability.
> We have updated the main text accordingly to improve clarity.
>
>
> **Q9: Algorithm 1 should be included in the main paper.**
>
> **A for Q9:**
> Thank you for the valuable feedback.
> As suggested, we have included Algorithm 1 in the main paper.

---

> ### Author Response · Authors · 2025-11-21
> **Response to Reviewer tmQR: Part 4**
>
> **Q10: More discussion on the guidance signal smoothing technique, and cases of the guidance choice make the agent hard to learn.**
>
> **A for Q10:**
> Thank you for the valuable feedback.
> Both the quality distribution of the guidance signal and the scheduling of guidance signal inclusion are essential for effective skill learning.
> We provide a detailed explanation below, and have expanded on these points in Appendix F as suggested.
>
> **(1) Quality distribution of the guidance signal.**
> The quality distribution of the guidance signal is mainly impacted by the distribution of labeled states, as our guidance signal construction method relies on the nearest neighbor approach.
> Effective skill learning requires high-quality guidance signals to sufficiently cover the state space, necessitating a well-scattered distribution of labeled states.
> Apart from encouraging sufficient coverage directly (through entropy-based sampling during query selection, as is used in COMPASS), we also explored alternative strategies for distributing high-quality guidance signals.
> Specifically, we considered reducing the uncertainty of the guidance signal using uncertainty-based sampling (Uncertainty), and adopting a random sampling approach (Uniform).
> As shown in Table 3 of the main paper (reproduced below), both the Uncertainty and the Uniform method perform significantly worse than COMPASS, showing the significance of sufficient coverage of high-quality guidance signals across the state space for effective skill learning.
>
> | Method | Ant Hole | Ant North Ant | Range |
> | :-: | :-: | :-: | :-: |
> | **COMPASS** | **1149.20 ± 127.05** | **1333.20 ± 129.10** | **329.80 ± 110.41** |
> | Uniform | 633.80 ± 279.41 | 1257.80 ± 189.01 | 40.80 ± 130.20 |
> | Uncertainty | 1034.40 ± 468.49 | 1059.20 ± 79.16 | -79.60 ± 302.86 |
>
> **(2) Scheduling of guidance signal inclusion.**
> For the scheduling of guidance signal inclusion, gradually introducing the guidance signal into the USD is crucial for effective skill learning.
> We have tried to abruptly introduce the guidance signal into the USD. However, this significantly degraded the performance.
> As shown in Table 4 (reproduced below), pure GSD performs the worst.
> This is primarily because the algorithm employs USD to gather data during early training, which distributes skills across the state space. However, when USD is abruptly switched to GSD, some skills become trapped within human-undesirable areas, where they receive zero rewards and are almost impossible to escape with the limited exploration mechanism in DSD.
>
> | $k_\beta$ | Ant Hole | Ant North | Ant Range |
> | :-: | :-: | :-: | :-: |
> | 3 | 1041.60 ± 226.01 | 1349.00 ± 162.01 | 312.40 ± 59.77 |
> | 5 | 1149.20 ± 127.05 | 1333.20 ± 129.10 | 362.20 ± 94.55 |
> | 10 | 1068.40 ± 321.47 | 1251.80 ± 99.38 | 399.40 ± 65.12 |
> | $\infty$ (no smoothing) | 991.60 ± 188.15 | 1120.80 ± 420.01 | 320.00 ± 91.07 |
>
> **Q13: More GSD baselines.**
>
> **A for Q13:**
> Thank you for your valuable suggestion to include additional GSD baselines.
> As suggested, we conduct additional experiments comparing with [2], which learns a reward model from pairwise human comparisons to guide the skill discovery process.
> As shown in the table below, COMPASS outperforms [2], primarily due to its ability to handle sparse labels more effectively.
> While [2] struggles to learn from sparse labels, COMPASS achieves superior performance by leveraging a semantically coherent skill latent space.
> This approach provides a dense, training-free guidance signal that is both robust and efficient, eliminating the need for auxiliary model training.
>
> | Method | Ant North | Ant Range |
> | :-: | :-: | :-: |
> | **COMPASS** | **1333.20 ± 129.10** | **362.20 ± 94.55** |
> | $[2]$ | -39.20 ± 14.88 | 72.40 ± 1.96 |

---

> ### Author Response · Authors · 2025-11-21
> **Response to Reviewer tmQR: Part 5**
>
> **Q14: Analysis of the dead zones, and experiments with a greater number of skills.**
>
> **A for Q14:**
> We thank the reviewer for the insightful comment on under-explored dead regions, such as areas behind hole hazards in the Ant Hole task.
>
> (1) As suggested, we conducted additional experiments by increasing the skill latent dimension from 2 to 4, thereby enriching the representational capacity for diverse behaviors.
> As visualized in Fig. 8 (Appendix E), the expanded skill space did not lead to exploration of the dead zones.
>
> (2) We believe this limitation primarily stems from the inherent exploration mechanism of the underlying DSD framework (e.g., METRA), rather than limitations of the guidance function $w(s)$.
> As shown in Fig. 2, both COMPASS and DoDont fail to explore areas behind the holes in the Ant Hole task.
> Even with accurate guidance, as demonstrated by the ``Oracle'' results, the base skills fail to bypass obstacles and reach areas hidden behind them.
>
> We use Fig. 9 (Appendix F) to illustrate that in scenarios with obstacles, METRA's optimal behaviors do not encourage skills to explore regions behind obstacles.
> Consider a mass point in a ``Hole'' scenario, with a 2-dimensional skill space, skills are expected to move in diverse directions.
> If we ignore the human-undesirable regions, the METRA's latent space $\phi(s)$ aligns with the 2D state space, i.e., $\phi(s)=s$.
> Since COMPASS's objective (Eq. 10) differs from METRA only by reweighting, if the samples to train the latent space are sufficient and the non-human-undesirable regions are well explored, the latent space COMPASS learned will be equivalent to that METRA learned in the non-human-undesirable regions.
>
> Consequently, for the specific skill latent $z$ shown in Fig. 9 in Appendix F, the optimal behavior follows the blue trajectory, which achieves the largest reward $(\phi(s')-\phi(s))^T z$ within the fixed timesteps.
> The trajectory of such optimal behavior will go parallel with the skill latent $z$ after passing the human-undesirable regions, which makes the region just behind the hole under-explored.
>
> The main reason for the dead zone is that METRA uses the DSD framework only to encourage the coverage of skills in the state space by aligning the trajectory with the uniformly distributed skill latent $z$, without using any exploration schemes such as prediction errors, state entropy, or pseudo-counts.
> Integrating the DSD framework with pure exploration strategies holds promise for mitigating the issue discussed above.
> We sincerely appreciate the reviewer's insightful comments and plan to investigate this direction in future work.
>
>
>
> **References:**
>
> [1] Do's and Don'ts: Learning Desirable Skills with Instruction Videos. NeurIPS 2024.
>
> [2] Controlled Diversity with Preference: Towards Learning a Diverse Set of Desired Skills. AAMAS 2023.

---

> ### Author Response · Authors · 2025-11-27
> **Looking forward to further discussions!**
>
> Dear reviewer,
>
> We were wondering if our response and revision have cleared all your concerns. In the previous response, we have tried to address all the points you have raised. We would appreciate it if you could kindly let us know whether you have any other questions, so that we can still have time to respond and address. We are looking forward to discussions that can further improve our current manuscript. Thanks!
>
> Best regards,
>
> The Authors

---

### Official Review · Reviewer_LQ8d · 2025-10-28

**Soundness:** 3
**Presentation:** 3
**Contribution:** 2
**Rating:** 6
**Confidence:** 4

**Summary:**

COMPASS addresses the pretraining phase of unsupervised skill discovery. Specifically, the authors assume there are some “good” pretraining behaviors to learn and “bad” ones to avoid. For example, a good behavior could be learning different skills that move north-ward, whereas a bad behavior may be moving in the other directions. Notably, we desire imbuing these biases during the skill pretraining phase, and not the downstream finetuning phase, because we would like to explore useful skills given these constraints and human priors.

To do this, COMPASS augments an existing guided skill discovery framework, DSD / Do’s Don’t, with (1) a semantically coherent latent space and (2) human-in-the-loop queries. A coherent latent space is a prerequisite for this type of constrained skill discovery, and the human-in-the-loop queries are necessary as the only way of identifying good/bad transitions.

**Strengths:**

1. The problem is scoped appropriately.
2. COMPASS results are compelling and show an improvement over existing baselines.

**Weaknesses:**

1. The contributions are fairly minimal. First, the scope of this is limited to simulated environments with scripted teacher. Second of all, the contribution on top of DoDon’t is minimal, with the primary differences being the different scoping.
2. I think it could be made slightly clearer what parts of the method are strictly taken from prior work, and which parts are novelties for this method.
3. Prior work on human feedback does not feel adequately represented in related work / prior works.
4. Some of the numbers (Table 3, 4, etc.) are kind of hard to parse. For example, could the state space coverage be normalized so it reports what % of the trajectories (or steps within the trajectory) are taken in the safe zone? Essentially normalizing the existing numbers -- if I understand how it is being computed.

**Questions:**

1. How important is the semantic latent space?
2. I understand that the goal is learning useful pretrained skills. However, since there is some metric of desired downstream tasks (“move in general direction,” “avoid these hazards”), why not incorporate this, so the skill discovery reward supplements a pretraining task reward? Baselines where DIAYN/METRA are aware of pretraining hazards/rewards could be valid baselines.

---

> ### Author Response · Authors · 2025-11-21
> **Response to Reviewer LQ8d: Part 1**
>
> Dear Reviewer,
>
> We sincerely thank you for your thoughtful and constructive feedback, as well as your positive assessment of the paper's soundness, presentation, and strong empirical results.
> We are grateful for your valuable comments, which have greatly helped us enhance the quality of our manuscript.
> We hope the following statement clears your concern.
>
>
>
>
> **W1-2: scope and contributions.**
>
> **A for W1-2**:
>
> (1) **Contribution beyond DoDont:**
> We appreciate your attention to the contribution of our work.
> Below, we clarify the distinctions and advancements of COMPASS compared to DoDont.
>
> COMPASS shares two similarities with DoDont:
>
> 1. It employs the same form of Guided Skill Discovery (GSD) objective, using a guidance signal $w(s)$ to reweight the original DSD objective.
> 2. It is built on the same DSD framework (METRA) as DoDont.
>
> However, COMPASS contributes beyond DoDont in the following key aspects:
>
> 1. We propose a **training-free method for constructing guidance signals**. Leveraging the semantically coherent latent space, the constructed guidance signal is theoretically proven to be reliable with minimal labels, without the need for auxiliary model training.
> In contrast, DoDont requires learning a guidance signal model before skill discovery, which is time-consuming and prone to overfitting on limited data, thus being unreliable.
> 2. We provide a **detailed analysis to justify adopting the METRA framework**, showing the necessity of a semantically coherent latent space for training-free guidance.
> This analysis not only supports our use of METRA's temporal distance but also allows for selecting other semantically coherent DSD methods beyond METRA, ensuring future extensibility.
> In contrast, DoDont directly adopt METRA without further analysis.
> 3. We propose an **active query selection method** to gather human feedback, enabling adaptive selection of high-quality data during skill training.
> In contrast, DoDont relies on pre-collected human-labeled trajectories, making its performance heavily dependent on dataset quality, posing challenges in constructing such datasets.
>
> We thank you for your detailed comment and hope this response clarifies the unique contributions of our work.
>
>
>
> (2) **Scope beyond scripted teachers:**
> To demonstrate that COMPASS extends beyond scripted teacher scenarios, we conduct additional experiments involving real human labelers.
> Specifically, we engage human labelers to provide feedback on visualized 2D trajectories in the Ant Range task, following task descriptions.
> We conduct five runs with different seeds, collecting 40 human labels per run.
> As shown in the table below and visualized in Fig. 7 (Appendix E), COMPASS consistently achieved superior safe state coverages, confirming its practical effectiveness with real human input.
>
>
> | | Ant Range |
> | :-: | :-: |
> | COMPASS (oracle teacher) | 362.20 ± 94.55 |
> | COMPASS (human labelers) | 361.40 ± 49.95 |
>
>
>
> **W3: More discussion on prior work on human feedback.**
>
> **A for W3:**
> As suggested, we have expanded Section 5 to discuss [1] and [2], which utilize human feedback via pairwise comparisons to train a reward model for guiding skill discovery. We have also included a discussion on the broader historical context of human feedback in policy learning in Appendix F, as follows:
>
> > (Section 5) ... Recent studies explore utilizing pairwise human preferences [1, 2] to learn a human-aligned reward model. These models then guide the skill discovery process by identifying preferred regions [1] or encouraging alignment between skills and human values [2]. Despite these advancements, deriving expert trajectories or constraint formulas is often challenging and impractical in complex tasks, and classifiers or reward models trained on limited data can be unstable. This paper aims to address these limitations.
>
> > (Appendix F) **Human Feedback in Policy Learning**. Prior works have demonstrated the effectiveness of integrating human feedback into policy learning to overcome the challenges of manual reward design. Early works such as Daniel et al. (2014) used active queries with numerical ratings, ...
> Since feedback is scarce in our work, and the pretraining phase involves numerous potential tasks, which makes the indistinguishability issue more severe, COMPASS adopts discrete ratings for single segments, similar to (Akrour et al., 2014) and (Sugiyama et al., 2012), to avoid the indistinguishability problem inherent in pairwise comparisons. Additionally, unlike standard approaches that require training parameterized reward models, COMPASS proposes a training-free method that efficiently utilizes sparse feedback by leveraging the semantic coherence of the unsupervised skill latent space.

---

> ### Author Response · Authors · 2025-11-21
> **Response to Reviewer LQ8d: Part 2**
>
> **W4: Normalized safe state coverage results.**
>
> **A for W4:**
> We appreciate the reviewer's insightful suggestion that normalizing state space coverage can improve clarity.
> As suggested, we present the normalized safe state coverage results in the table below (also shown in Table 6 in Appendix E).
> COMPASS also outperforms most baselines on this metric, demonstrating that most of its visit states are safe and human-desired.
>
> | **Method**  | **Ant North**    | **Ant Range**    | **Ant Hole**      | **HalfCheetah Right** |
> | ----------- | ---------------- | ---------------- | ----------------- | --------------------- |
> | **Oracle**  | 92.60 ± 1.60     | 94.30 ± 2.00     | 98.90 ± 0.70      | 99.00 ± 0.00          |
> | DIAYN       | 0.00 ± 0.00    | 0.00 ± 0.00      | **100.00 ± 0.00** | 50.00 ± 0.00          |
> | LSD         | 18.30 ± 10.90    | 36.40 ± 26.10    | 69.00 ± 13.50     | 28.50 ± 4.20          |
> | METRA       | 20.40 ± 14.70    | 23.90 ± 2.10     | 74.70 ± 2.20      | 48.00 ± 1.00          |
> | DoDont      | 93.40 ± 4.70     | 36.70 ± 5.70     | 83.20 ± 4.50      | 86.70 ± 5.30          |
> | **COMPASS** | **96.90 ± 2.50** | **80.90 ± 7.80** | 90.60 ± 2.40      | **98.70 ± 0.50**      |
> | | | | |
> |   | **Quadruped North** | **Humanoid Hole** | **Ant Range-North** | **Ant Hole-North** |
> | **Oracle**  | 79.80 ± 2.90        | 91.20 ± 5.30      | 95.80 ± 1.50        | 98.60 ± 0.30       |
> | DIAYN       | 6.20 ± 8.50         | **100.00 ± 0.00** | 0.00 ± 0.00         | 100.00 ± 0.00      |
> | LSD         | 20.60 ± 24.20       | **100.00 ± 0.00** | 36.40 ± 26.10       | 74.50 ± 1.50       |
> | METRA       | 21.10 ± 10.60       | 82.30 ± 15.40     | 23.90 ± 2.10        | 75.60 ± 2.50       |
> | DoDont      | 76.00 ± 8.00        | 84.20 ± 4.30      | 41.30 ± 6.20        | 83.90 ± 5.00       |
> | **COMPASS** | **87.60 ± 7.50**    | 90.50 ± 9.80      | **81.40 ± 8.50**    | **93.10 ± 4.80**   |
> | | | | |
> |   | **HalfCheetah Not-Flip** | **Safety-Gym Hazard** |   |   |
> | **Oracle**  | 100.00 ± 0.00   | 37.00 ± 4.70          |   |   |
> | DIAYN | 75.20 ± 14.00    | 28.30 ± 7.20          |   |   |
> | LSD | 76.10 ± 5.80 | 23.20 ± 7.80          |   |   |
> | METRA | 89.20 ± 3.20 | 28.30 ± 9.30 |   |   |
> | DoDont | 91.70 ± 2.80 | 26.50 ± 8.30 |   |   |
> | **COMPASS** | **100.00 ± 0.00**        | **40.00 ± 6.70**      |   |   |
>
> **Q1: Importance of the semantically coherent latent space.**
>
> **A for Q1**:
> A semantically coherent latent space is crucial for training-free guidance, as it ensures that states with nearby embeddings share similar human desirability, allows sparse human labels to effectively propagate across the state space, forming a dense guidance signal.
>
> To validate its importance, we conduct additional experiments using the Euclidean distance between raw states as the distance constraint in the DSD framework, that is,
> $$
> ||\phi(x) - \phi(y)||_2 \le ||x-y||_2, \, \forall x,y \in \mathcal{S}
> $$
> As shown in the table below, using Euclidean distance performs worse than COMPASS, which uses the semantically coherent latent space.
> This highlights the importance of semantically coherent latent representations.
>
> | | Ant North | HalfCheetah Not-Flip |
> | :-: | :-: | :-: |
> | **COMPASS** | **1333.20 ± 129.10** | **215.60 ± 3.05** |
> | Euclidean distance | 657.60 ± 432.54 | 104.20 ± 21.51 |
>
> **Q2: Hazard-aware rewards as baselines.**
>
> **A for Q2:**
> We appreciate this insightful suggestion and agree on it as a valuable baseline.
> We are pleased to confirm that this aligns precisely with our "Oracle" baseline, as it achieves hazard-aware skill discovery by multiplying a perfect guidance signal with the intrinsic reward of skill discovery (i.e., METRA).
> The perfect guidance signals reflect the suggested downstream tasks, such as "move in a specific direction" or "avoid certain hazards".
> The "Oracle" baseline serves as a performance upper bound, illustrating the potential when human intent is perfectly and explicitly defined.
>
> In practice, however, explicitly specifying human intent as a reward function is often challenging [3, 4].
> The core contribution of our work lies in deriving the required guidance signal $w(s)$ in a training-free manner, using only sparse and potentially noisy human feedback.
> As shown in Tables 1, 2, and 6, COMPASS achieves performance comparable to the "Oracle", indicating that our guidance mechanism effectively captures human intent.
>
> We sincerely thank the reviewer again for the timely and valuable comments. We hope our responses adequately address your concerns.
>
> **References:**
>
> [1] Controlled Diversity with Preference: Towards Learning a Diverse Set of Desired Skills. AAMAS 2023.
>
> [2] Human-Aligned Skill Discovery: Balancing Behaviour Exploration and Alignment. AAMAS 2025.
>
> [3] PEBBLE: Feedback-Efficient Interactive Reinforcement Learning via Relabeling Experience and Unsupervised Pre-training. ICML 2021.
>
> [4] Preference Transformer: Modeling Human Preferences using Transformers for RL. ICLR 2023.

---

> ### Author Response · Authors · 2025-11-27
> **Looking forward to further discussions!**
>
> Dear reviewer,
>
> We were wondering if our response and revision have cleared all your concerns. In the previous response, we have tried to address all the points you have raised. We would appreciate it if you could kindly let us know whether you have any other questions, so that we can still have time to respond and address. We are looking forward to discussions that can further improve our current manuscript. Thanks!
>
> Best regards,
>
> The Authors

---

> > ### Comment · Reviewer_LQ8d · 2025-11-27
> > **Thank you for the response!**
> >
> > Thank you for addressing my concerns. I plan on keeping my score!

---

### Official Review · Reviewer_AmYS · 2025-10-31

**Soundness:** 2
**Presentation:** 3
**Contribution:** 2
**Rating:** 6
**Confidence:** 4

**Summary:**

This paper proposes a guided skill discovery method as an extension to METRA where it learns an extra weight function based on human feedback to reduce intrinsic reward in undesired states. The weight function is based on the nearest neighbor method and thus does not require parameter learning. Additionally, the authors propose an active learning method to gather human feedback, based on state coverage.

**Strengths:**

1. The paper is well written. The clear presentation and the adequate coverage of preliminaries make the paper easy to follow.
2. The proposed framework of combining reward weight in the representation space learned by METRA is interesting and novel.
3. The ablation of human label query methods is helpful for evaluating the effectiveness of different active learning methods.

**Weaknesses:**

My major concerns are the effectiveness of the **non-parametric** guidance function $w(s)$ and the assumption that $\phi$ learned by METRA is a good representation space to **semantically** adjust skill behavior.

Specifically, the paper mainly focuses on navigation tasks with relatively simple unsafe zones. However, in many tasks the unsafe space would be complex. For example, in the kitchen environment used by METRA, we would like the arm to avoid self-collision and collision with other fixed furniture. In the configuration space, the unsafe region would be a complex non-convex space, and I doubt the proposed non-parametric form of guidance function could effectively representation such a space. Especially as current experiment results already somewhat show COMPASS's ineffectiveness with slightly complex safe/unsafe boundaries: in all hole environments, the skills learned by COMPASS, though avoiding the whole, barely explores the region behind the hole. Instead, most skills simply keep moving straight forward after passing the holes. Of course, given that METRA mainly learns moving straight skills, it's hard to tell whether the under-exploration comes from METRA's limitation of $w$'s ineffectiveness. Any insights / empirical results on how the proposed non-parametric $w$ can (or how to adapt it to) work / scale to complex safety boundaries are welcome.

**Questions:**

1. L440, "Boundary" approach should be "Uncertainty"?

---

> ### Author Response · Authors · 2025-11-21
> **Response to Reviewer AmYS**
>
> Dear Reviewer,
>
> We sincerely thank you for your insightful and constructive feedback, as well as your positive assessment of our method's novelty and clarity.
> We are grateful for your valuable comments, which have helped us enhance the quality of our manuscript.
> We hope the following statement clears your concern.
>
>
> **W1: Effectiveness of the non-parametric guidance $w(s)$ to complex safety boundaries.**
>
> **A for W1:**
> We thank the reviewer for raising this crucial point regarding the scalability of our non-parametric guidance function $w(s)$ to complex, non-convex unsafe regions.
>
> (1) **Effectiveness in non-trivial safety settings**:
> While many of our experiments involve navigation tasks, we would like to highlight **COMPASS's strong performance in tasks with more complex safety constraints**.
> In the HalfCheetah Not-Flip task, COMPASS successfully guides the agent to avoid a dynamic behavioral hazard (flipping), achieving 100\% safe state ratio and near-oracle state coverage (Tables 1 and 6).
> These results validate $w(s)$'s capacity to handle non-geometric and behaviorally complex constraints.
>
>
> (2) **Under-exploration in "Hole" tasks**:
> We agree with your keen observation that skills in Ant Hole environments tend to under-explore regions behind the holes.
> As shown in Fig. 2, both COMPASS and DoDont fail to explore areas behind the holes in the Ant Hole task.
> Even with accurate guidance, as demonstrated by the ``Oracle'' results, the base skills fail to bypass obstacles and reach areas hidden behind them.
>
> We believe this is primarily due to the inherent exploration mechanisms of the underlying METRA framework.
> In scenarios with obstacles, METRA's optimal behaviors do not encourage skills to explore regions behind obstacles.
> We use Fig. 9 (Appendix F) to illustrate this phenomenon.
> Consider a mass point in a ``Hole'' scenario, with a 2-dimensional skill space, skills are expected to move in diverse directions.
> If we ignore the human-undesirable regions, the METRA's latent space $\phi(s)$ aligns with the 2D state space, i.e., $\phi(s)=s$.
> Since COMPASS's objective (Eq. 10) differs from METRA only by reweighting, if the samples to train the latent space are sufficient and the non-human-undesirable regions are well explored, the latent space COMPASS learned will be equivalent to that METRA learned in the non-human-undesirable regions.
>
> Consequently, for the specific skill latent $z$ shown in Fig. 9 in Appendix F, the optimal behavior follows the blue trajectory, which achieves the largest reward $(\phi(s')-\phi(s))^T z$ within the fixed timesteps.
> The trajectory of such optimal behavior will go parallel with the skill latent $z$ after passing the human-undesirable regions, which makes the region just behind the hole under-explored by any skills.
>
> The main reason for the dead zone is that METRA uses the DSD framework only to encourage the coverage of skills in the state space by aligning the trajectory with the uniformly distributed skill latent $z$, without using any exploration schemes such as prediction errors, state entropy, or pseudo-counts.
>
> (3) **Scale to complex safety boundaries**:
> We fully agree that scaling to highly non-convex and structured unsafe spaces requires further advancements.
> As discussed above, integrating the DSD framework with pure exploration strategies holds promise for mitigating the issue discussed above.
> We sincerely appreciate the reviewer's insightful comments and plan to investigate this direction in future work.
>
>
>
>
> **Q1 "Boundary" should be "Uncertainty" in L440:**
>
> **A for Q1:**
> We thank the reviewer for pointing out this and have corrected it in the revised manuscript.
>
>
> We sincerely thank the reviewer again for the timely and valuable comments. We hope our responses adequately address your concerns.

---

> > ### Comment · Reviewer_AmYS · 2025-11-26
> >
> > I appreciate the detailed explanation from the authors. All my concerns are resolved. I view the paper positively as before and will keep my score the same.

---

### Official Review · Reviewer_wg5W · 2025-11-02

**Soundness:** 2
**Presentation:** 4
**Contribution:** 2
**Rating:** 2
**Confidence:** 4

**Summary:**

The paper presents a way to achieve guided skill discovery without relying on predefined rules, expert demonstrations or training instruction models. The method is based on  the insight that behaviours with a similar human desirability lie close to each other in a semantically coherent skill latent space. This insight is used to construct a training-free guidance signal which is then used in skill discovery in order to learn aligned and safe skills with minimal human feedback.

**Strengths:**

The paper addresses an important problem – that of learning aligned skills via skill discovery. The overall method is quite intuitive and it seems to be effective. The paper is also generally presented well.

**Weaknesses:**

Although the approach seems simple and intuitive, I am not very convinced about the way human labels are used. I suspect treating all states within a segment to have the same label may require segments to be very short, and may not work well in general.

Also, I am unsure how COMPASS compares to other existing approaches that aim to solve the same problem. For instance, two works that seem very related, but are not discussed in the paper are:
[1] Controlled Diversity with Preference: Towards Learning a Diverse Set of Desired Skills. In Proceedings of the 2023 International Conference on Autonomous Agents and Multiagent Systems. 2023.

[2] Human-Aligned Skill Discovery: Balancing Behaviour Exploration and Alignment. In Proceedings of the 24th International Conference on Autonomous Agents and Multiagent Systems (AAMAS '25). International Foundation for Autonomous Agents and Multiagent Systems, Richland, SC, 1025–1033.

Particularly [1], which learns a desired state region from human preferences seems to have quite a lot of similarities with the paper, perhaps enough to warrant an empirical comparison, at least in the simple environments in Fig 2.

**Questions:**

1. The assumption that all states within segment carry the same label seems overly simplistic. I would assume this approach only works when very small segments are used. Have the authors considered using approaches like diverse density [3] to better capture the landscape of ‘good’, ‘bad’ and ‘neutral’ states?

2. Related to the previous question – how does the performance vary with longer trajectory segments?

3. What happens to states that are part of both ‘good’ as well as ‘bad’ trajectory segments? How are the labels for these states decided?

4. How many samples does COMPASS use for different types of problems?

5. I would also be curious to seem how sensitive COMPASS would be with decreasing availability of human data.

6. How sensitive is COMPASS’s performance with and without active query selection?

[3] "A framework for multiple-instance learning." Advances in neural information processing systems (1997).

---

> ### Author Response · Authors · 2025-11-21
> **Response to Reviewer wg5W: Part 1**
>
> Dear Reviewer,
>
> We sincerely thank you for your insightful and constructive feedback, as well as your positive assessment of our method's intuitiveness and clarity.
> We are grateful for your valuable comments, which have helped us enhance the quality of our manuscript.
> We hope the following statement clears your concern.
>
>
>
>
> **W1, Q1, Q2: Human label usage and robustness to segment length.**
>
> **A for W1, Q1, Q2:**
> We sincerely thank the reviewer for the valuable feedback.
>
> (1) **Robustness of COMPASS to segment lengths $H$:**
> As suggested, we conducted additional experiments to evaluate COMPASS with varying segment lengths $H$. As shown in the table below and visualized in Fig. 4 (Appendix E), COMPASS consistently achieves superior safe state coverage across different segment lengths ($H=20, 40, 60$), demonstrating its robustness to this parameter.
> The segment length $H=60$ is comparable to SOTA PbRL works (use segment length 50) [1, 2, 3] and the suggested work (use segment length 50) [5], which we believe is not excessively short.
>
> | $H$  |     Ant North     |   Ant Range    |
> | :--: | :---------------: | :------------: |
> |  20  | 1333.20 ± 129.10  | 362.20 ± 94.55 |
> |  40  | 1325.20 ±   75.70 | 309.25 ± 77.00 |
> |  60  | 1327.60 ± 132.60  | 347.00 ± 35.24 |
>
>
> (2) **Applicability of diverse density[4]:**
> We greatly appreciate your insightful suggestion to use Diverse Density (DD) [4] for modeling "good/neutral/bad" states.
>
> - **Similarities**: We agree that DD or multi-instance learning has some similarities to our work. Specifically, DD (or other multi-instance learning methods) addresses uncertainty in instance-level labels by utilizing bag-level annotations, which align conceptually with COMPASS's use of segment-level human feedback.
> Furthermore, the computation of COMPASS's guidance signal $w(s)$ (Eq. 7), which considers the distance between states in the latent space and the annotated state set, is conceptually similar to the distance-weighted probability modeling in DD's noisy-or method.
> - **Limitations**: However, DD has limitations in directly addressing the three-class problem (good/neutral/bad).
> Also, its reliance on concept points to model probabilities assumes clustering of positive and negative samples [7], and requires a predefined number of concept points, which may not suit tasks like Ant Hole in our study.
> Moreover, DD and related multi-label learning (MIL) methods involve optimization processes, which may contradict COMPASS's key advantage of avoiding auxiliary model training.
> - **Summary**: Given that these challenges could not be easily addressed, we did not adopt DD in our current work.
> Nevertheless, given their similarities, we believe DD and MIL offer alternative perspectives for understanding our method, as well as a promising aspect for further analysis and improvement.
> To this end, we have incorporated discussions on DD and MIL in Appendix F to contextualize their relevance to our approach.
>
>
>
> **W2: Discussion on related works [5, 6].**
>
> **A for W2:**
> We sincerely thank the reviewer for highlighting these highly relevant and excellent works. We agree that [5, 6] address critical aspects of guided skill discovery and have incorporated them into Section 5, providing a more detailed discussion and comparison with these papers.
>
> In summary, COMPASS differs from [5, 6] in several key aspects:
> - **Our contributions**: Our primary contribution lies in the **training-free guidance method** for Guided Skill Discovery (GSD), which is not demonstrated by prior works, including [5, 6].
> This is achieved through a key insight: a **semantically coherent skill latent space** enables the use of minimal "good/neutral/bad" labels to directly construct a dense guidance signal, without additional training.
> We believe this insight, along with the entire training-free framework, represents a novel and meaningful contribution to GSD.
> - **Differences**: Additionally, our setting differs from [5, 6], as we use direct segment labeling instead of pairwise comparisons, and require significantly fewer labels (40 or 100) than [5, 6], which use over 1000 labels. Also, [6] requires human evaluation of state-action sequences, which is more challenging than our focus on state sequences alone.
>
> As suggested, we conduct additional experiments comparing [5] with COMPASS.
> As shown in the table below, while [5] outperforms baselines without guidance and is comparable to DoDont, it does not surpass COMPASS. This demonstrates the superior performance of COMPASS against strong baselines.
>
> |   Method    |      Ant North       |     Ant Range      |
> | :---------: | :------------------: | :----------------: |
> | **COMPASS** | **1333.20 ± 129.10** | **362.20 ± 94.55** |
> |   CDP [5]   |    -39.20 ± 14.88    |    72.40 ± 1.96    |
> |   DoDont    |   1307.20 ± 188.33   |  -427.60 ± 224.09  |
> |     LSD     |  -1056.80 ± 515.24   |  -916.80 ± 589.79  |
> |    METRA    |  -1425.80 ± 756.14   | -1247.40 ± 147.97  |

---

> ### Author Response · Authors · 2025-11-21
> **Response to Reviewer wg5W: Part 2**
>
> **Q3: Handling states in both "good" and "bad" segments.**
>
> **A for Q3:**
> We sincerely thank the reviewer for the insightful question.
> First, the state spaces in our tasks are continuous, making it highly unlikely to sample identical states.
> If a state $s_0$ appears in both "good" and "bad" segments, states $s$ within a small neighborhood of $s_0$ in the latent space will have similar distances to both the positive and negative labeled state sets, i.e., $d_\phi(s, D_0) \approx d_\phi(s, D_2)$.
> As a result, the softmax-based guidance signal $w(s)$ naturally assigns a value close to 1, treating the state as "neutral", neither desirable nor undesirable.
> This aligns with human intuition, as ambiguous states should not strongly affect exploration.
>
>
>
> **Q4: Number of samples across tasks.**
>
> **A for Q4:**
> COMPASS uses 40 labeled segments for state-based tasks (Ant, HalfCheetah, Safety-Gym) and 100 labeled segments for pixel-based tasks (Quadruped, Humanoid), as specified in Appendix H.3.
>
>
>
> **Q5: Performance under decreasing availability of human data.**
>
> **A for Q5:**
> As suggested, we conduct two additional experiments to evaluate COMPASS (1) with a reduced number of samples and (2) under noisy labeling conditions.
>
> (1) **Evaluation under fewer samples:**
> We evaluate COMPASS with a reduced number of samples. As shown in the table below and visualized in Fig. 5 (Appendix E), COMPASS effectively aligns with human intent even with only 10 or 20 labels, while its performance improves as the number of labels increases.
>
> | # of labels |    Ant North     |     Ant Hole     |
> | :---------: | :--------------: | :--------------: |
> |     40      | 1333.20 ± 129.10 | 1149.20 ± 127.05 |
> |     20      | 1035.60 ± 591.51 | 829.60 ± 243.73  |
> |     10      | 801.40 ± 654.11  | 766.20 ± 220.39  |
>
>
> (2) **Evaluation under noisy labeling conditions:**
> To simulate human labeling errors, we randomly assign labels to states within a band of width $R_\text{error}$ around the safety boundaries, reflecting potential human uncertainty in these regions.
> As shown in the table below and visualized in Fig. 6 (Appendix E), COMPASS remains robust under noisy labels, indicating the reliability of our training-free guidance mechanism.
>
>
> | $R_\text{error}$ |    Ant North     |   Ant Range    |
> | :--------------: | :--------------: | :------------: |
> |        0         | 1333.20 ± 129.10 | 362.20 ± 94.55 |
> |       0.5        | 1184.60 ± 124.84 | 369.40 ± 51.74 |
> |        1         | 1084.20 ± 135.81 | 360.80 ± 43.53 |
>
>
>
>
>
> **Q6: Performance with and without active query selection.**
>
> **A for Q6:**
> As shown in Table 3 (in the main paper) and reproduced below, COMPASS with our active query selection consistently outperforms uniform sampling (which is **without active query selection**) and uncertainty-based methods, demonstrating that our active query selection effectively improves performance by maximizing the coverage of labeled states in the state space.
>
>
> |   Method    |       Ant Hole       |      Ant North       |      Ant Range      |
> | :---------: | :------------------: | :------------------: | :-----------------: |
> | **COMPASS** | **1149.20 ± 127.05** | **1333.20 ± 129.10** | **329.80 ± 110.41** |
> |   Uniform   |   633.80 ± 279.41    |   1257.80 ± 189.01   |   40.80 ± 130.20    |
> | Uncertainty |   1034.40 ± 468.49   |   1059.20 ± 79.16    |   -79.60 ± 302.86   |
>
>
>
> **References:**
>
> [1] Pebble: Feedback-efficient interactive reinforcement learning via relabeling experience and unsupervised pre-training. ICML 2021.
>
> [2] RIME: Robust Preference-based Reinforcement Learning with Noisy Preferences. ICML 2024.
>
> [3] Query-Policy Misalignment in Preference-Based Reinforcement Learning. ICLR 2023.
>
> [4] A Framework for Multiple-Instance Learning. NeurIPS 1997.
>
> [5] Controlled Diversity with Preference: Towards Learning a Diverse Set of Desired Skills. AAMAS 2023.
>
> [6] Human-Aligned Skill Discovery: Balancing Behaviour Exploration and Alignment. AAMAS 2025.
>
> [7] Multiple Instance Learning: A Survey of Problem Characteristics and Applications. Pattern Recognition, 2018.

---

> ### Author Response · Authors · 2025-11-27
> **Looking forward to further discussions!**
>
> Dear reviewer,
>
> We were wondering if our response and revision have cleared all your concerns. In the previous response, we have tried to address all the points you have raised. We would appreciate it if you could kindly let us know whether you have any other questions, so that we can still have time to respond and address. We are looking forward to discussions that can further improve our current manuscript. Thanks!
>
> Best regards,
>
> The Authors

---

> > ### Comment · Reviewer_wg5W · 2025-11-27
> > **Thanks for the response!**
> >
> > I thank the authors for their detailed responses, and appreciate the efforts made to address all the reviewers' comments. Most of my questions have been addressed. I have a few follow-up questions/comments regarding the use of human feedback:
> >
> > 1. The method proposed here seems to be learning from an order of magnitude fewer feedback signals compared to other methods, such as the ones using human preferences on trajectory segments. What is fundamentally responsible for this?
> >
> > 2. Noisy variants of preference-based RL can be simulated using BPref [1] models to simulate the various ways in which human feedback may be noisy. Although you have demonstrated a way in which noise can be added to labels, would you concede that perhaps BPref and preference-based RL offers a more structured manner of introducing noise?
> >
> > [1] Lee, Kimin, et al. "B-Pref: Benchmarking Preference-Based Reinforcement Learning."

---

> > > ### Author Response · Authors · 2025-11-28
> > >
> > > Dear Reviewer,
> > >
> > > We are glad that our previous response addressed most of your concerns.
> > > We also appreciate the valuable comments, which helped us significantly enhance the quality of our manuscript. We hope the following statement clears your remaining concern.
> > >
> > > **Q1: The reason why the method proposed here seems to be learning from an order of magnitude fewer feedback signals compared to other methods.**
> > >
> > > **A for Q1:**
> > > Thank you for the insightful question regarding COMPASS's sample efficiency.
> > > We believe the superior performance of COMPASS stems from leveraging the semantical coherence latent space to effectively propagate human feedback.
> > >
> > > Specifically, COMPASS significantly enhances sample efficiency by leveraging a semantically coherent latent space, where nearby states in the latent space share similar human desirability.
> > > This latent space is trained with a large amount of unlabeled data, which ensures its reliability.
> > > This allows the influence of individual good/neutral/bad labels to propagate across neighboring regions via a distance-weighted softmax (Eq. 7), making it fundamentally more efficient than methods that treat each label independently.
> > >
> > > **Q2: BPref and preference-based RL offer a more structured manner of introducing noise.**
> > >
> > > **A for Q2:**
> > > We appreciate the reviewer's insightful comments.
> > > We agree that BPref [1] offers a structured framework for incorporating noise in pairwise comparison PbRL.
> > > However, its specific noisy variants (Stoc, Skip, Equal, Myopic) are inherently designed for pairwise settings.
> > > These methods rely on the Bradley-Terry model to introduce noise by skipping queries, labeling queries as equal, or accounting for differences in oracle rewards between two segments.
> > > Consequently, these mechanisms are not directly applicable to our segment-label-based setting.
> > >
> > > On the other hand, our noise design is more general, applicable to both pairwise query-based PbRL and our segment-label-based method.
> > > Specifically, we introduce noise at the boundary between good and bad regions, a strategy consistent with prior studies [2, 3] and the Mistake variant of BPref, further validating our approach.
> > > For future work, we plan to explore more structured noise mechanisms that extend beyond BPref, aiming for effectiveness across both pairwise and segment-label settings.
> > >
> > >
> > > We sincerely thank the reviewer again for the thoughtful and constructive feedback. We hope that our responses and additional experimental results have addressed your concerns.
> > >
> > >
> > >
> > > **References:**
> > >
> > > [1] B-Pref: Benchmarking Preference-Based Reinforcement Learning. NeurIPS 2021.
> > >
> > > [2] S-EPOA: Overcoming the Indistinguishability of Segments with Skill-Driven Preference-Based Reinforcement Learning. IJCAI 2025.
> > >
> > > [3] RIME: Robust Preference-based Reinforcement Learning with Noisy Preferences. ICML 2024.

---

### Meta-Review · Area_Chair_Kq5Z · 2026-01-06

**Summary:**

This work proposes COMPASS, a guided skill discovery method. The core idea is to leverage human feedback to avoid undesired states. The topic itself is interesting and well motivated. During the rebuttal period, the authors addressed several reviewer concerns effectively, including questions about underlying assumptions and the addition of relevant baselines.

However, since the method is developed on top of existing prior work, the novelty and overall contribution are not fully convincing. More broadly, while the authors provided responses to concerns related to theoretical aspects and real human experiments, not all issues were fully resolved during the rebuttal.

Overall, this is a solid and interesting piece of research, and with further refinement it could achieve a strong outcome in a future submission. In its current version, however, it still falls somewhat short.

**Reviewer Concerns:**

### Reviewer wg5W

* [resolved] Assumptions on human labels: An ablation study and additional justification were provided.
* [resolved] Comparison with prior work: CDP [5] was added.
* [resolved] Feedback efficiency

### Reviewer AmYS

* [resolved] Effectiveness of the non-parametric guidance

### Reviewer LQ8d

* [partially] Limited contribution: Clarified through additional discussionㄴ.

* [resolved] related work

### Reviewer tmQR

* [partially resolved] Evaluation under practical human feedback signals: Real human experiments were included; however, they were conducted in relatively simple environments. More thorough evaluations with additional baselines would be necessary.

* [remaining] Theoretical concerns

* [resolved] Formal description of guided skill discovery

* [resolved] Explanation for the optimization problem transformation.

* [partially] term/notion

**Reviewer Scores:**

* Reviewer AmYS, Reviewer LQ8d: keeping original score (6)

* Reviewer wg5W: 2 $\rightarrow$ (4 or 6)

* Reviewer tmQR: 2 $\rightarrow$ (2 or 4)

---

### Decision · Program_Chairs · 2026-01-26

Reject